# STRUCTURED JOINT ALEATORIC AND EPISTEMIC UNCERTAINTY FOR HIGH DIMENSIONAL OUTPUT SPACES

## ABSTRACT

Uncertainty estimation plays a vital role in enhancing the reliability of deep learning model predictions, especially in scenarios with high-dimensional output spaces. This paper addresses the dual nature of uncertainty — aleatoric and epistemic — focusing on their joint integration in high-dimensional regression tasks. We introduce an approach to approximate joint uncertainty using a low-rank plus diagonal covariance matrix, which preserves essential output correlations while mitigating the computational complexity associated with full covariance matrices. Specifically, our method reduces memory usage and enhances sampling efficiency and log-likelihood calculations. Simultaneously, our representation matches the true posterior better than factorized joint distributions, offering a clear advancement in reliability and explainability for deep learning model predictions. Furthermore, we empirically show that our method can efficiently enhance out of distribution detection in specific applications.

## 1 INTRODUCTION

In the realm of deep learning, uncertainty estimation plays a pivotal role in enhancing the reliability of model predictions. This paper delves into the domain of uncertainty estimation, specifically focusing on regression tasks in high-dimensional output spaces.

In these scenarios, model predictions exhibit two types of uncertainty: aleatoric and epistemic Kendall & Gal (2017). Heteroscedastic aleatoric or data uncertainty can be modeled as an inherent component of the model output. There, the model output consists of the parametrization of an assumed distribution, such as a Gaussian distribution in the case of regression or a categorical distribution in the case of classification. This distribution is learned through minimizing its negative log-likelihood. In contrast, due to its complexity in deep neural networks, epistemic or model uncertainty is typically approximated by sampling from an proxy distribution of models Hüllermeier & Waegeman (2021).

Combining both in a single model usually results in a so-called second-order distribution Bengs et al. (2023). On the one hand, it consists of a distribution over model weights capturing epistemic uncertainty. On the other hand, it models a distribution over plausible predictions representing aleatoric uncertainty. Sampling from the model weights and performing a transformation (forward pass) of the input data results in another distribution representing the aleatoric uncertainty. The shape of this second-order distribution limits further analysis, as it is difficult to visualize, and it does not allow for calculation of the marginal likelihood of a sample. Therefore, the second-order distribution is typically marginalized and approximated by a single distribution, representing the joint uncertainty.

Traditionally, these uncertainties have been jointly modeled without considering correlations between outputs (e.g. pixels), assuming independent factorized univariate Gaussian distributions Kendall & Gal (2017). However, neglecting correlations can limit the comprehensive understanding of uncertainties, especially in scenarios where dependencies between model outputs exist, such as in pixel-wise semantic segmentation Monteiro et al. (2020), pixel-wise regression tasks, such as optical flow estimation or image in-painting, or graph node regression. Figure 1 illustrates the increased representational power of full covariance matrices (right) compared to merely diagonal ones (left). In both cases, samples from the weight space lead to multiple predictions consisting of mean and covariance each. The expected covariance is used for calculating the aleatoric component $\Sigma^a$, whereas the covariance of the means is used to calculate the epistemic component $\Sigma^e$. The sum of both results in the joint covariance matrix $\Sigma^a + \Sigma^e = \Sigma$.

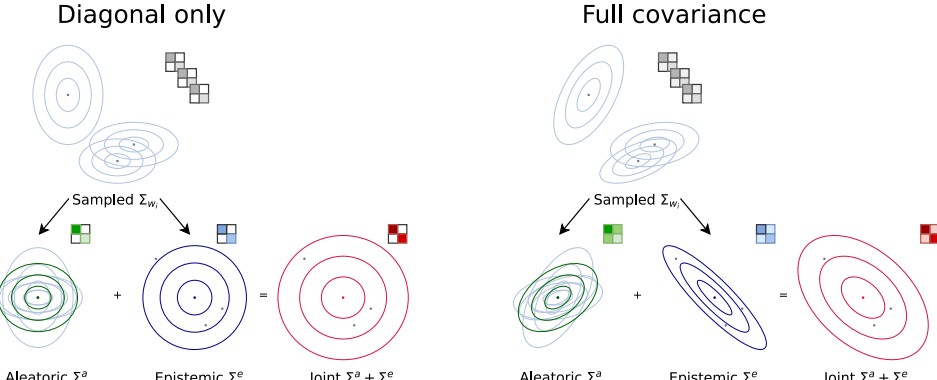

Figure 1: Visualization of covariance matrices for the 2D case. Three samples with corresponding means and covariances are depicted (light blue). The columns show the inferred aleatoric (green), epistemic (blue) and joint uncertainty (red), respectively. On the left, the covariance matrices are purely diagonal, limiting their representational power. To the right, the same matrices are depicted with non-diagonal values kept, allowing them to capture the overall uncertainty in greater detail.

Yet, incorporating correlations in high-dimensional output spaces poses a significant challenge, given that the number of correlations between output dimensions scales quadratically in terms of memory complexity $\mathcal{O}(S^2)$ with the total number of outputs $S$. This leads to large covariance matrices, requiring considerable storage space and making calculations computationally infeasible. This renders many downstream operations like sampling from a normal distribution parameterized by these covariance matrices, which involves Cholesky decomposition $\mathcal{O}(S^3)$, computing the log-likelihood of samples with matrix inversion $\mathcal{O}(S^3)$ and determinant computation (e.g. $\mathcal{O}(S^3)$ with lower-upper (LU) Decomposition) practically impossible.

In summary, an efficient representation of the joint uncertainty containing aleatoric and epistemic uncertainty without neglecting covariances for high-dimensional output spaces has not yet been explored.

**Related Work**  To estimate epistemic uncertainty, various Bayesian frameworks have been developed, including methods like stochastic variational inference Blundell et al. (2015), Monte Carlo dropout Gal & Ghahramani (2016), deep ensembles Lakshminarayanan et al. (2017), stochastic weight averaging Maddox et al. (2019), or Laplace approximation Daxberger et al. (2021). The modeling of heteroscedastic aleatoric uncertainty has been well-established for some time Nix & Weigend (1994); Skafte et al. (2019); Stirn & Knowles (2020); Seitzer et al. (2022). Building upon these works, others have unified epistemic and aleatoric uncertainty in a single model Kendall & Gal (2017); Depeweg et al. (2018); Stirn et al. (2023); Immer et al. (2024). However, all aforementioned methods either evaluate their method only for prediction tasks with a single output value or approximate the marginalized likelihood as a factorized Gaussian, disregarding inter-pixel correlations.

Covariances for uncertainty estimation have been modeled in various applications, including localization Russell & Reale (2021), human pose estimation Gundavarapu et al. (2019), pixel regression Dorta et al. (2018a;b); Duff et al. (2023), multi class predictions Willette et al. (2021), and segmentation Monteiro et al. (2020). Some approaches that predict full covariance matrices are limited to low dimensional model output spaces Russell & Reale (2021); Gundavarapu et al. (2019). Approaches for handling high-dimensional output spaces typically sparsify the covariance matrix. However, some of these approaches can only model uncertainty in the local neighborhood using a band Cholesky parametrization Dorta et al. (2018a;b); Duff et al. (2023). Some works Salinas et al. (2019); Monteiro et al. (2020); Willette et al. (2021); Nussbaum et al. (2022) use a low-rank plus diagonal (LR+D) parametrization, which is capable of capturing global correlations. Nehme et al. (2024); Yair et al. (2024) learn the low-rank factors of aleatoric uncertainty directly without adding a diagonal and create a rank-deficient semi-definite covariance matrix. This may be sufficient for both sampling and analysis, but it does not provide the positive definiteness required for calculating the log-likelihood. Importantly, all these sparse solutions merely focus on aleatoric uncertainty. Zepf et al. (2023) combine aleatoric and epistemic uncertainty with a LR+D representation. However, by partially using the Maximum a posteriori (MAP) solution as a further approximation, they do not account for the influence of the model uncertainty on the estimation of the aleatoric uncertainty, leading overall to a worse uncertainty estimate. Furthermore, they do not resolve the second-order distribution to provide

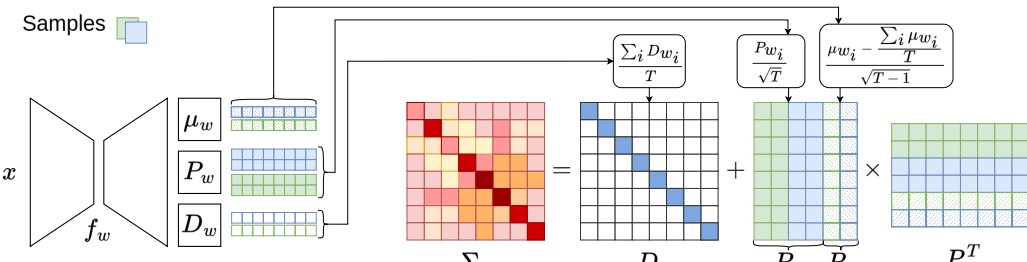

Figure 2: Construction of our LR+D matrix. A network predicts values $\mu_w$, $P_w$, and $D_w$ for two exemplarily sampled weights $w_i$, respectively (green and blue). By averaging and stacking these values in a specific manner, we build the diagonal $D$ and the low-rank matrix $P$ as parts of our LR+D representation of $\Sigma$. See Section 2 for an in-depth explanation.

a joint representation suitable for further analysis, such as log likelihood calculation, and its usage is limited to consecutive sampling.

In conclusion, while significant advancements have been made in modeling covariances for uncertainty estimation, the existing approaches suffer from limitations such as local sparsification, inadequate joint representations, and neglect of weight space uncertainty, indicating a need for further research to develop more comprehensive and globally accurate uncertainty estimation methods.

**Contribution** This work is the first to efficiently combine both aleatoric and epistemic uncertainties within any high-dimensional sparse joint representation, leveraging the LR+D framework. Unlike existing approaches that approximate the second-order distribution with a factorized normal distribution, neglecting correlations between outputs, our method maintains crucial correlations between outputs while avoiding the heavy space and time requirements of a full covariance matrix even for high-dimensional output spaces. We showcase the superior representational power of our approach on multiple high-dimensional regression tasks, e.g. inpainting on the MNIST dataset, colorization of grey-scale versions of the CelebA celebrity faces dataset Liu et al. (2015), and optical flow estimation on the Flying Chairs dataset Dosovitskiy et al. (2015).

## 2 METHOD

In this paper, we consider supervised learning tasks with high dimensional output spaces $y \in \mathbb{R}^S$, with $S$ denoting the number of output units, e.g. pixels times the number of output channels. We aim to approximate the posterior distribution $p(W|\boldsymbol{X}, \boldsymbol{Y})$ over model weights $W$ given input-output data pairs $(\boldsymbol{X}, \boldsymbol{Y})$. As computing $p(W|\boldsymbol{X}, \boldsymbol{Y})$ directly is generally infeasible for neural networks, we approximate it using Bayesian methods like Monte Carlo Dropout (MCD), Stochastic Variational Inference (SVI), Deep Ensemble (DE) to provide a proxy distribution $q_\theta^*(W)$, parametrized by $\theta$.

To represent the joint uncertainty, we decompose the posterior predictive distribution $p(y|x, \boldsymbol{X}, \boldsymbol{Y})$ of an unseen input-output pair $(x, y)$ into two terms: $p(y|x, W)$, representing the likelihood of the output given the input $x$ and the network weights $W$, and the posterior distribution of weights given the data $p(W|\boldsymbol{X}, \boldsymbol{Y})$. We model $p(y|x, W)$ as a multivariate Gaussian distribution $p(y|x, W) = \mathcal{N}(\mu_W(x), \Sigma_W(x))$, where we keep the spatial complexity of the covariance matrix $\Sigma_W(x)$ low by constructing it in LR+D form. That is, we formulate it as a sum of small matrices, $\Sigma_W(x) = D_W(x) + P_W(x)P_W^\mathsf{T}(x)$, with $D_W$ denoting a diagonal matrix of shape $S \times S$ and $P_W$ a tall matrix of shape $S \times R^W$. We choose a rank $R^W$ much lower than the number of outputs $R^W \ll S$, such that only the most important directions of the aleatoric covariance are covered. We further enforce $D_W$ to contain strictly positive diagonal entries and since $P_W P_W^\mathsf{T}$ is always symmetric, $\Sigma_W$ is always symmetric positive definite by construction and thus a valid covariance matrix. The ultimate goal of this work is to calculate an efficient yet representative representation of the posterior predictive distribution $p(y|x, \boldsymbol{X}, \boldsymbol{Y})$.

### 2.1 MODELING THE JOINT UNCERTAINTY

We start modeling the parameters of the posterior predictive distribution consisting of mean and covariance by using Monte Carlo integration to approximate the expected model output $\mathbb{E}[y|x, \boldsymbol{X}, \boldsymbol{Y}] \approx \mu(x)$. The empirical mean is given as $\mu(x) = \frac{1}{T}\sum_i^T \mu_{w_i}(x)$, where $T$ repre-

sents the number of weight samples drawn from $w_i \sim q_\theta^\star(W)$. The joint covariance matrix can be split into epistemic and aleatoric uncertainty using the law of total variance as

$$\underbrace{\mathrm{Cov}\left[y|x, \boldsymbol{X}, \boldsymbol{Y}\right]}_{\underset{\text{joint uncertainty}}{\widetilde{\Sigma}(x)}} \approx \underbrace{\mathrm{Cov}_{q_\theta^*(W)}\left[\mu_W(x)\right]}_{\underset{\text{epistemic uncertainty}}{\widetilde{\Sigma}^e(x)}} + \underbrace{\mathbb{E}_{q_\theta^*(W)}\left[\Sigma_W(x)\right]}_{\underset{\text{aleatoric uncertainty}}{\widetilde{\Sigma}^a(x)}}. \tag{1}$$

This suggests that the mean of covariance matrices across forward pass samples captures aleatoric uncertainty, whereas the covariance of the means represents epistemic uncertainty. We provide a complete derivation of equation 1 in the supplement D.4.

Our objective is to represent the joint uncertainty $\Sigma(x)$ in LR+D form as the sum of aleatoric and epistemic uncertainties,

$$D + PP^\intercal = (D^e + P^e P^{e\intercal}) + (D^a + P^a P^{a\intercal}), \tag{2}$$

where $D^a$, $D^e$, and $D$ are diagonal matrices and $P^a$, $P^e$, and $P$ low-rank matrices representing aleatoric, epistemic, and joint uncertainties, respectively. Then, $D = D^e + D^a$ and $P = [P^a \quad P^e]$, where $[\quad]$ denotes columnwise block concatenation. This expression allows us to conveniently represent both aleatoric and epistemic uncertainties in LR+D form, simplifying further analysis and computation. Figure 2 provides an intuitive illustration about the construction of our LR+D matrix components. Starting with $\Sigma^e$, we describe in detail the individual components of our LR+D representations in the following sections.

## 2.2 Epistemic Uncertainty

The epistemic uncertainty is estimated through the distribution over weights. To derive its covariance, we employ empirical sampling from the proxy distribution over model weights as follows:

$$\Sigma^e(x) = \frac{1}{T-1} \sum_i^T \left(\mu_{w_i}(x) - \mu(x)\right)\left(\mu_{w_i}(x) - \mu(x)\right)^\intercal \quad w_i \sim q_\theta^*(W) \tag{3}$$

Our objective is to avoid the full covariance matrix and instead seek a representation in LR+D form.

**Naive Representation** To bring the approximated epistemic covariance matrix into LR+D form, we set the diagonal $D^e(x)$ to zero and rewrite the covariance matrix as $\Sigma^e(x) = P^e(x)P^e(x)^\intercal$, where $P^e(x) \in \mathbb{R}^{S \times R^e}$ has $R^e = T$ columns and is defined as

$$P^e(x) = \frac{1}{\sqrt{T-1}} \left[\mu_{w_1}(x) - \mu(x) \quad ... \quad \mu_{w_T}(x) - \mu(x)\right]. \tag{4}$$

In high dimensional scenarios, the number of samples will often be much lower than the number of outputs $T \ll S$ rendering $\Sigma^e$ singular and therefore non-invertible. Computing enough samples for a full rank estimate is usually prohibitive with regard to time and space complexity. In general, one can expect more accurate results from increased sample sizes. However, in this naive representation, larger sample sizes also result in quadratic scaling of computational complexity. Hence, we suggest further approximations to cope with moderately high sample sizes.

**Truncated Singular Value Decomposition Approximation** Assuming that samples are often correlated and exhibit dominant directions of variance, we propose to reduce the dimensionality of $P^e(x)$ with truncated Singular Value Decomposition (SVD). Keeping only the most informative columns of $P^e(x)$ will improve the efficiency of further computations without losing much information. However, the calculation of SVD comes with its own computational complexity that has to be taken into account. As we use the same nomenclature for further parts of the model, we omit the superscript $()^e$ and the dependency on $(x)$ for this section. Specifically, we decompose the matrix $P$ as $P^\intercal = U\Psi V^\intercal$, where $U$ and $V$ are orthogonal matrices, and $\Psi$ is a diagonal matrix containing the singular values in non-decreasing order $\Psi_{1,1} \leq ... \leq \Psi_{S,S}$. Subsequently, we define the matrix $\tilde{P} = V\Psi$ and rewrite the matrix product as $\Sigma = PP^\intercal = \tilde{P}\tilde{P}^\intercal$. To reduce dimensionality, we discard the smallest singular values and their associated columns in $V$. However, we keep the univariate variance parts of these dropped columns by transferring them to a new diagonal matrix $\hat{D}$. Hence,

the approximated matrix $\hat{\Sigma} = \hat{D} + \hat{P}\hat{P}^\intercal$ keeps all independent variance and the most important covariances of $\Sigma$. If we keep the $\hat{R}$ largest singular values, the components of $\hat{\Sigma}$ are

$$\hat{P} = \begin{bmatrix} V_{R-\hat{R}} \cdot \Psi_{R-\hat{R},R-\hat{R}} & ... & V_R \cdot \Psi_{R,R} \end{bmatrix} \tag{5}$$

and

$$\hat{D}_{ii} = \sum_{j=1}^{R-\hat{R}-1} V_{ij}^2 \cdot \Psi_{j,j}^2. \tag{6}$$

The number of columns to keep has to be chosen empirically. The aforementioned approach enables us to effectively represent epistemic uncertainty in the LR+D form.

## 2.3 ALEATORIC UNCERTAINTY

Similar to epistemic uncertainty, the covariance matrix capturing aleatoric uncertainty $\Sigma^a(x)$ can be approximated through empirical sampling. We calculate the empirical mean of covariance matrix estimations over all sampled model weights via

$$\Sigma^a(x) = \frac{1}{T} \sum_{i}^{T} \Sigma_{w_i}(x) \quad w_i \sim q_\theta^*(W). \tag{7}$$

We here again intend to represent $\Sigma^a(x)$ in LR+D form.

**Naive Representation**  To rewrite the covariance matrix containing the aleatoric uncertainty in LR+D representation, we reformulate $\Sigma^a(x) = D^a(x) + P^a(x)P^a(x)^\intercal$ using

$$D^a(x) = \frac{1}{T} \sum_{i}^{T} D_{w_i}(x) \tag{8}$$

$$P^a(x) = \frac{1}{\sqrt{T}} \begin{bmatrix} P_{w_1}(x) & ... & P_{w_T}(x) \end{bmatrix}. \tag{9}$$

This yields a $P^a \in \mathbb{R}^{S \times (T \cdot R^W)}$ with $T \cdot R^W$ columns. Although $T \cdot R^W$ generally remains far below $S$, $P^a$ can still become fairly large as the number of drawn samples increases. Thus, we further reduce the number of columns of $\Sigma^a$ as for the epistemic case.

**Truncated SVD Approximation**  Like in the previous Subsection 2.2, we reduce the dimensionality of $P^a$ via SVD. This leads for $\hat{P}^a$ to the same equation as 5. For the diagonal matrix, we need to incorporate both, the average calculated in 8 and the removed variance by the SVD given by 6, so the resulting diagonal is given by:

$$\hat{D^a}_{ii} = D_{ii}^a + \sum_{j=1}^{R-\hat{R}-1} V_{ij}^2 \cdot \Psi_{j,j}^2. \tag{10}$$

## 3 EXPERIMENTS

### 3.1 EXPERIMENTAL SETUP

**Proposed Method**  We empirically evaluate our method of joint aleatoric and epistemic uncertainty modeling using our LR+D representation in several experiments. In all experiments, we use variants of the U-Net Ronneberger et al. (2015) architecture. We adapt the U-Net for Bayesian inference by adding dropout, which we use for MCD Gal & Ghahramani (2016) and DE Lakshminarayanan et al. (2017) or by using variational convolutional layer for SVI Blundell et al. (2015) to estimate a distribution over model weights which estimates epistemic uncertainty. However, we note that our approach is compatible with any Bayesian method suitable for large model outputs. We use a single model with multiple outputs for the mean and the covariance prediction parts. To train this model, we gradually we train the mean separately and gradually change increase the weight on the log likelihood loss for the covariance parameters. For architectural details, please refer to our code in the repository.

**Datasets and Tasks** We evaluate our method in different settings on the MNIST, CelebA, and Flying Chairs datasets for the tasks of inpainting, colorization, and optical flow.

We train a reconstruction model to inpaint distorted handwritten digits from the MNIST dataset. For the inpainting task, we mask out $5/7$ of the image area. We use the official test set and split the training set into 50,000 train and 10,000 validation images. Figure 3.2 (bottom) shows reconstruction results for a single digit.

To evaluate performance on optical flow estimation, we use the Flying Chairs Dosovitskiy et al. (2015) dataset. This dataset is resized to 192 x 256 and split into 18,297/2,287/2,288 training/validation/test images. We provide visualizations of the predictions as part of the supplement.

To evaluate our method on facial images, we use the CelebA CelebA-HQ dataset, keeping the original splits from CelebA Liu et al. (2015). The original split contains 24,183 images for training, 2,993 for validation, and 2,824 for testing (image size 256 × 256). We study two tasks on this dataset: colorization and inpainting.

**Baselines** We compare the performance of non-Bayesian models with different Bayesian approaches. Additionally, we incorporate a diagonal (D) covariance matrix following the method of Kendall & Gal (2017). We extend this by using a low-rank plus diagonal (LR+D) parameterized distribution.

As a further baseline, we also provide results by following the approach by Zepf et al. (2023). Here, we further approximate the aleatoric uncertainty term of Equation 1 to prevent sampling aleatoric $P$ matrices. This reduces the number of resulting columns from $T \times (R + 1)$ to $T + R$. It is achieved by approximating the expectation of the aleatoric uncertainty $\Sigma^a$ term with the aleatoric covariance prediction of the model with the expected weights:

$$\Sigma^a = \mathbb{E}_{q_\theta^*(W)} \left[ \Sigma_W(x) \right] \approx \Sigma_{\mathbb{E}_{q^*}[W]}(x)$$

To compute this term, we require the expected weights of the Bayesian models to be well-defined. For MCD, this is done by turning dropout off and rescaling the activations accordingly. For SVI, where the weights follow Gaussian distributions, the expected weights are simply the means of the Gaussian distributions. For DE, we are unable to define expected weights, hence this approximation is not evaluated in this case. Note that Zepf et al. (2023) refer to this approach as MAP solution, which coincides with the expected weights solution if the weight uncertainty is modeled with symmetrical unimodal distributions like Gaussians as commonly used by SVI and Laplace Approximation (LA). Furthermore, Zepf et al. (2023) do not provide a joint representation, and log likelihood calculation is only possible using a combination of our methods.

**Hyperparameter** Finally, we evaluate our joint the LR+D parametrization in combination with all three Bayesian methods. For this case, we let the model predict a matrix $P_W \in \mathbb{R}^{S \times R^W}$ of rank $R^W = 8$ and for epistemic models, we draw $T = 64$ samples. The predictions are multivariate Normal distributions, represented by their LR+D parametrization. Those predictions are joined to a single, LR+D parametrized distribution. For the full joint uncertainty LR+D model, this yields a joint $P$ matrix with $R = T \times (R^W + 1) = 576$ columns, which we jointly compress down to $R = 64$ with truncated SVD while keeping the diagonal variance of the dropped columns as described in Section 2. For the expected weights baseline, we perform an additional forward pass using the expected weights and concatenate the aleatoric and epistemic columns, which leads to $R = 72$ in total. All models are trained for the same amount of steps.

## 3.2 MAIN RESULTS - COMPARISON OF THE FIT OF PREDICTIVE DISTRIBUTIONS

**Quantitative Results** To evaluate performance, we use the negative log-likelihood, which measures how well a model predicts the observed data. Lower values imply that the model's predictions are closer to the actual outcomes. Quantitatively, we find that modeling epistemic uncertainty improves the likelihood of unseen test sample predictions, as shown in Table 1. This finding holds for both modeling the diagonal and modeling the LR+D across all experiments. Additionally, including covariances through our LR+D further improves the likelihood of unseen test sample predictions across all experiments. The usefulness of expected weights $\mathbb{E}[W]$ approximation for the aleatoric uncertainty $\Sigma^a$ is inconsistent. Our approach, to combine both into a joint multivariate uncertainty representation and using SVD, is superior in all performed experiments with all tested Bayesian methods.

| Epistemic | Parameters | | MNIST Inpainting ×1 | CelebA Colorization ×1000 | Inpainting ×100 | Flying Chairs Optical Flow ×100 |
|---|---|---|---|---|---|---|
| ✗ | D | - | 2665± 7794 | -146± 111 | 153± 175 | 1059± 496 |
| ✗ | LR+D | - | 2610±24982 | -216± 78 | 134± 70 | 882± 554 |
| MCD | D | - | -292± 345 | -152± 77 | 125± 146 | 1012± 390 |
| SVI | D | - | -268± 297 | -157± 35 | 125± 79 | 1043± 381 |
| DE | D | - | -308± 236 | -158± 58 | 93± 101 | 965± 291 |
| MCD | LR+D | $\mathbb{E}[W]$ | -155± 4257 | -235± 41 | 73± 62 | 853± 342 |
| SVI | LR+D | $\mathbb{E}[W]$ | -327± 3232 | -225± 38 | 116± 35 | 760± 524 |
| MCD | LR+D | SVD | **-409**± 302 | **-241**± 28 | 66± 52 | 844± 308 |
| SVI | LR+D | SVD | -383± 2591 | -229± 31 | 109± 31 | **748**± 467 |
| DE | LR+D | SVD | -394± 118 | -240± 25 | **61**± 44 | 837± 272 |

Table 1: Quantitative Results. We evaluate the negative log-likelihoods (base 10) of predictions across various dataset-task combinations and its test set variability using standard deviations. Lower values indicate higher likelihood and, therefore, better predictions. Our method is assessed in four experiments: inpainting of removed image parts, colorization, and optical flow estimation. The log-likelihoods scale linearly with the dimensionality of the prediction and, in the case of masking, are evaluated only in the masked area. Results are generated using both Bayesian (MCD,SVD,DE) and non-Bayesian (✗) networks, each with diagonal (D) and low-rank plus diagonal (LR+D) covariance parametrization. For the combination of LR+D and Bayesian approaches, we depict both the results by expected weights $\mathbb{E}[W]$ approximation and by our approach using truncated SVD. Our findings demonstrate that employing the LR+D representation and incorporating epistemic uncertainty enhance the posterior predictive distribution and increase the likelihood of the predictions.

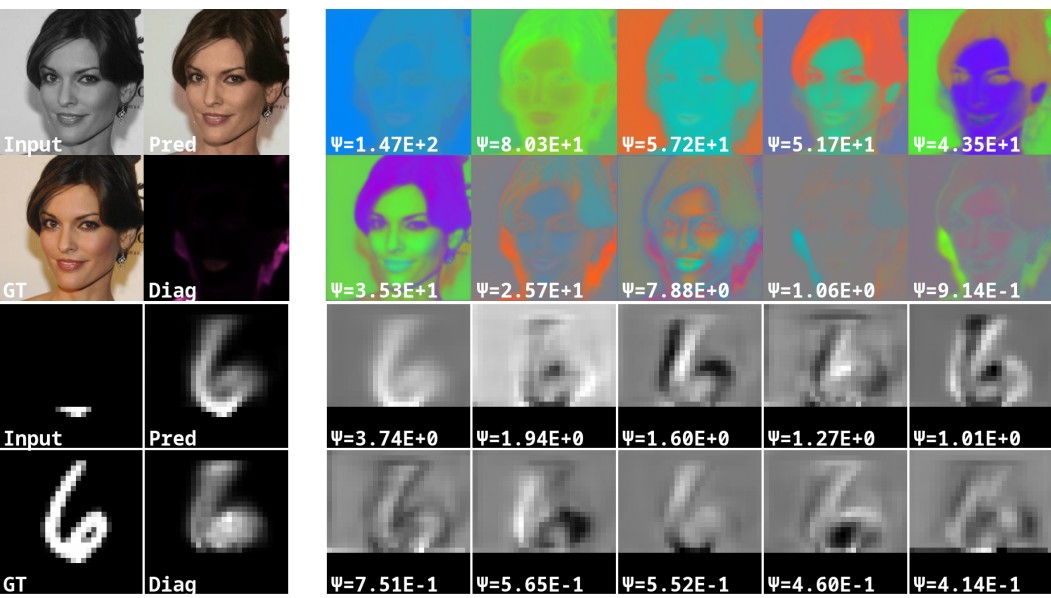

Figure 3: Qualitative Results. Random samples from the test sets depict the input, prediction, ground truth, and parameters of the predictive distribution. The top rows show colorization on CelebA images, while the bottom rows display inpainting of MNIST digits. Our model predicts a mean (Pred), the parameter $D$ (Diag), and a low-rank matrix $P$. In both cases, the predicted joint low-rank matrix $P$ is reduced to the 64 most significant directions (columns) based on their respective eigenvalues. The 10 images in columns 3-8 visualize the 10 most important directions with a random orientation in descending order of the associated eigenvalues. We observe that these columns focus on uncertainty in specific image areas or colors. Additionally, the singular values $\Psi$ measure the importance of the associated direction. For more qualitative results of all datasets and Bayesian methods, please see the Supplement Figures 6, 7, 8, 9,and 10.

**Qualitative Results** Figure 3 and Supplementary Figures 6, 7, 8, 9,and 10 provide qualitative results, where we exemplarily visualize how the 10 most important columns in our joint low-rank matrix $P$ describe the areas of correlated uncertainty. E.g. within the CelebA inpanting's (top) visualized $P$ Matrix, the first two eigenvectors focus on full image color shifts, whereas the first one focuses on an axis between orange and blue (inverse color), the second one on the axis between purple to green (inverse color). The third eigenvector focus on the color contrast between foreground and background. The fourth one focuses on hair and eye color. Further, the singular values $\Psi$ provide an interpretation of the importance of these correlations. Visualization of the eigenvectors is only possible with our method, which includes the covariance terms; hence, allowing the identification of image regions with correlated uncertainty. The Parameter D (Diag) captures additional uncertainty, which could not be captured by the Low Rank Covariance Matrix created by $PP^{\intercal}$.

In summary, these qualitative results can help to intuitively describe the underlying relations of uncertainty on an image level.

## 3.3 ADDITIONAL RESULT - OUT OF DISTRIBUTION DETECTION

We evaluate our method on the task of out-of-distribution detection on the MNIST dataset, where we omit the digit "2" from the training data. Similar to Subsection 3.2, we train four types of models, with and without epistemic uncertainty, using both diagonal and LR+D representations. Our aim is to study whether predictive distributions are more spread out for unseen out-of-distribution (OOD) test samples compared to in-distribution (ID) test samples. A common metric to evaluate the spread of continuous distributions is differential entropy Thomas & Joy (2006). Figure 4 presents histograms of differential entropies of the predictive distributions of two Bayesian models. The left one's output is parametrized using a diagonal normal distribution, whereas the right one uses the LR+D parametrization. We observe that the LR+D parametrized network better separates the ID entropies from the OOD entropies than the diagonally parameterized covariance model. For quantitative analysis, we compute the average entropy for ID and OOD samples in the respective columns. Furthermore, we calculate the Area Under the Curve (AUC) as a metric for the separation of OOD and ID samples using differential entropy. Finally, we employ a Kolmogorov–Smirnov (KS) test to measure a distance between the distribution of differential entropies of ID and OOD samples. Higher values are preferred for both metrics. Our results demonstrate that both 1) modeling epistemic uncertainty and 2) covariances help to improve the differentiation between ID and OOD test samples for the tested dataset. Finally, we present a combination of both, using the expected weights approximation $\mathbb{E}[W]$ and the SVD approximation. Our method, which combine both and compresses using SVD, is superior to all baselines.

## 3.4 SPATIAL AND COMPUTATIONAL COMPLEXITY

Figure 5 presents the memory (left) and time (right) requirements for computing the log-likelihood of different covariance parameterizations: sparse options like diagonal (D) and low-rank plus diagonal

| Epistemic | Parameters | | ID | OOD | AUC ↑ | KS↑ |
|---|---|---|---|---|---|---|
| ✗ | D | | -1134 | -988 | 0.782 | 0.441 |
| ✓ | D | | -751 | -561 | 0.851 | 0.564 |
| ✗ | LR+D | | -1407 | -1026 | 0.842 | 0.558 |
| ✓ | LR+D | $\mathbb{E}[W]$ | -1291 | -989 | 0.859 | 0.583 |
| ✓ | LR+D | SVD | -844 | -593 | **0.882** | **0.625** |

Table 2: Results for out-of-distribution (OOD) detection of MNIST digits using the LR+D representation. The digit "2" is excluded from the training set and, therefore, OOD in the test set. The table presents the results for a Bayesian and non-Bayesian network, both combined with D and with LR+D covariance parametrization. The columns in-distribution (ID) and OOD indicate the average differential entropy for the respective sample types. The column AUC measures the Area Under the Curve (AUC) for using differential entropy as a separation criterion. Finally, Kolmogorov–Smirnov (KS) depicts a distance between the distributions of ID and OOD differential entropies. Larger values are preferable for both metrics. The AUC as well as KS demonstrate that both epistemic uncertainty and the LR+D representation are beneficial for the detection of OOD samples.

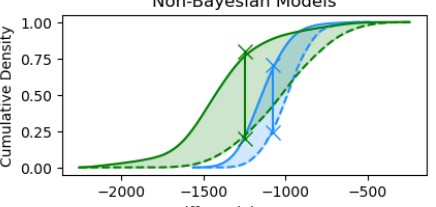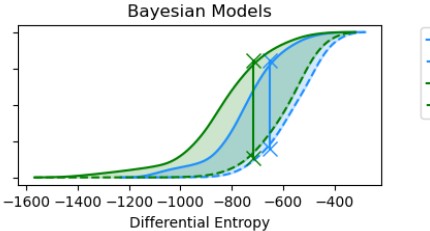

Figure 4: Distribution of differential entropies predicted by non-Bayesian and Bayesian models parametrized with either a D or an LR+D covariance matrix. The plots depict cumulative kernel-density plots of the differential entropies of all ID (solid line) and OOD (dashed line) test set samples. Using the LR+D representation, the ID and OOD samples show a better separation based on their entropy compared to using only the variances. The vertical line visualizes the result of the KS test.

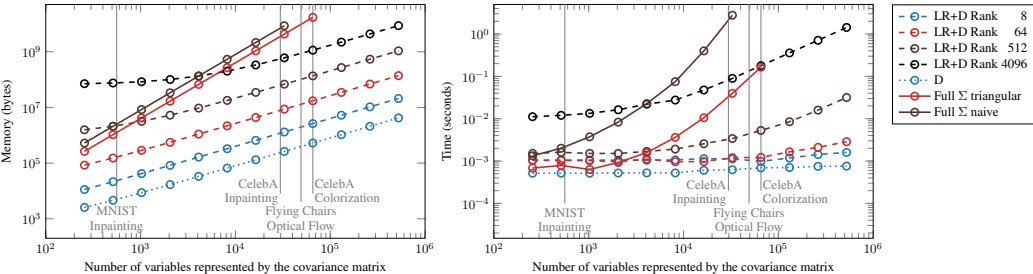

Figure 5: Empirical measurement of memory requirement of log likelihood calculation (left) and Average empirical time measurement of 100 log likelihood calculations (right). Memory requirements increase linear with the number of variables for LR+D representation and quadratic for full covariance representations. Full covariance matrix measurements are done until exceeding memory of the GPU. The LR+D representation enables handling larger covariance matrices than full rank parametrizations.

(LR+D), as well as full covariance $\Sigma$ using both naive and lower-triangular parameterizations. The complexity is shown as a function of the number of variables in the covariance matrix, with specific points marking the number of variables for each dataset-task combination.

For LR+D, we evaluate various numbers of columns $R$ in the low-rank matrix $P$. We limit our analysis to sizes that fit within a single 48GB GPU. As a result, the full covariance plot stops early when memory capacity is exceeded. As seen in the figure, the LR+D parameterization (with 64 columns) is significantly more efficient than the naive full covariance, both with respect to memory and time, for all datasets. In larger datasets like CelebA and Flying Chairs, the full covariance matrix approaches the GPU memory limit, even without batching or storing the model and its gradients. Theoretical details on the computational complexity can be found in the Supplementary Material B.

## 3.5 ABLATION STUDIES

For a comprehensive evaluation of our uncertainty framework, we carry out multiple ablations to study which factors are important for optimal model performance. One aspect is the number of samples drawn from the weight distribution and the number of columns $R$ retained in the final representation after performing SVD. We vary the number of samples $T$, the application of SVD to the low-rank matrix $P$, and the number of columns retained post-SVD, resulting in $R$ total columns. Table 3 presents the mean negative log-likelihood (NLL) and its test set variability using standard deviations. The results indicate that, generally, a higher number of samples and retained columns improve predictions. However, retaining a high number of columns resulting from sampling without dimensionality reduction can lead to numerical instabilities ($\star$, $\star\star$), preventing NLL calculation. In contrast, a low number of samples causes high variability within the test set. We find that optimal results are achieved through dimensionality reduction using SVD, balancing efficient representation with fewer columns against better predictive performance in terms of NLL.

We further ablate various design choices. In a first ablation we show that including the update of the diagonal given by the Equations 6 and 10 in Table 6 makes the predictions more robust. Further, we show better performance for the choice to perform SVD on the joint $P$ matrix instead of separated in $P^a$ and $P^e$ in Table 7; next we ablate on the number of columns of $P_w$ directly predicted by the

| | | | MNIST | CelebA | | Flying Chairs |
| | | | Inpainting | Colorization | Inpainting | Optical Flow |
| $T$ | $R$ | $P$ | $\times 1$ | $\times 1000$ | $\times 100$ | $\times 100$ |
|---|---|---|---|---|---|---|
| 64 | 576 | - | -455± 994 | ⋆⋆ | ⋆⋆ | ⋆793±274 |
| 32 | 288 | - | -440±1401 | ⋆-252±27 | 53±45 | ⋆813±289 |
| 16 | 144 | - | -411±2035 | -245±34 | 65±51 | 838±311 |
| 8 | 72 | - | -352±3096 | -237±45 | 81±59 | 868±341 |
| 64 | 64 | SVD(64) | -409± 302 | -242±35 | 68±51 | 844±308 |
| 64 | 32 | SVD(32) | -372± 204 | -235±41 | 82±57 | 866±322 |
| 64 | 16 | SVD(16) | -345± 170 | -228±45 | 96±63 | 888±336 |
| 64 | 8 | SVD( 8) | -319± 165 | -217±48 | 113±71 | 915±349 |

Table 3: Comparison of different approximations of the LR+D-parametrized covariance matrix using varying numbers of samples and degrees of dimensionality reduction via SVD. $T$ denotes the number of samples drawn from $q^\star(w)$, and $R$ indicates the number of columns in the resulting representation. The column $p$ specifies to what extent the dimensionality is reduced after sampling using SVD. Numbers in brackets represent the retained singular vectors, resulting in columns $R$. The columns to the right contain the mean negative log-likelihoods (NLL) and their test set variability using standard deviations. We indicate scaling to ease visualization. Some NLLs could not be evaluated due to numerical issues. Cells with more than 10% missing test results are replaced by ⋆⋆, those with at least one missing test result (<10%) are marked with ⋆. The results clearly show that an increasing number of samples improves performance (lower negative log-likelihood (NLL)) and consistency (lower standard deviation). However, it also increases the risk of numerical issues. To harness most of the performance benefits while avoiding numerical instabilities, we can apply SVD for dimensionality reduction. Additionally, retaining more columns after performing SVD results in better outcomes.

model weights in Table 8 and find a moderate improvement at higher computational cost motivating our choice of choosing 8 columns in further experiments. Finally, we provide an extensive list of various design choice combinations in the Ablation Table 9.

## 4 DISCUSSION

**Conclusion**   In this work, we have explored the dual nature of uncertainties — aleatoric and epistemic — and their integration in high-dimensional regression tasks. We proposed a novel method that employs a low-rank plus diagonal covariance matrix to approximate joint uncertainty, effectively preserving vital output correlations and significantly reducing the computational demands that are inherent to full covariance matrix representation. Our approach lowers memory usage and improves the efficiency of both sampling and log-likelihood calculations. Empirically, our approach outperforms the commonly used factorized Gaussian representation. It exhibits a lower negative log-likelihood, indicating superior performance in uncertainty estimation, particularly in high-dimensional regression tasks. Furthermore, it excelled in out-of-distribution (OOD) detection on the tested dataset, leveraging the criterion of differential entropy. This success underscores the method's effectiveness in capturing and quantifying uncertainty.

**Limitations**   Our method conceptually extends to any Bayesian framework; however, for simplicity and computational reasons, we restrict our evaluation to using Monte Carlo Dropout, Stochastic Variational Inference and Deep Ensemble. Further investigations into other Bayesian inference techniques should determine their empirical applicability. We expect that more advanced concepts will lead to better overall uncertainty estimation. The method is flexible with regard to the choice in number of columns utilized in the low-rank plus diagonal parameterization of the covariance matrix. A higher number of columns provides better overall uncertainty estimation but can lead to numerical instabilities and increased computational complexity. Essentially, our method exhibits a trade-off between the quality of uncertainty estimation and these factors, see Table 3. We hypothesize that the numerical instabilities are related to the linear dependencies of the columns in the low-rank and, hence, propose additional investigations to understand and mitigate their impact. Finally, our method builds upon the assumption that uncertainties in output can be modeled by a single multivariate Gaussian, even though this approximation is often used in the literature Kendall & Gal (2017); Monteiro et al. (2020); Duff et al. (2023). However, multivariate Gaussians may not be a suitable approximation for every task, for example, for uncertainties in translation or rotation in images. Exploring epistemic uncertainty under different distributions is a highly promising research question.

By more accurately approximating the true posterior than traditional joint distributions, our method enhances both the reliability and explainability of predictions from deep learning models.

## 5 REPRODUCIBILITY

The source code, released under an open-source license, is available via an anonymous public GitHub repository `https://anonymous.4open.science/r/structured_joint_uncertainty/`. Checkpoints for all models trained with the proposed method, as well as all baselines, are available upon request. The datasets used in the experiments are publicly accessible and links as well as preprocessing scripts are included in the repository. An extensive schematic and intuitive description of the method, along with proofs, is also included in the Supplementary Material. Additionally, qualitative examples are provided to enhance understanding of the method.

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

## A    SYMBOLS AND ACRONYMS

### A.1    LIST OF SYMBOLS

| Symbol | Remark |
|---|---|
| $D$ | Diagonal matrix used for LR+D |
| $P$ | Tall matrix $P$ used for LR+D |
| $R$ | Number of columns of tall matrix $P$ |
| $S$ | Number of outputs $P$ |
| $T$ | Number of epistemic samples |
| $U$ | Singular vectors |
| $\Psi$ | Diagonal matrix containing singular values of $P$ |
| $V$ | Eigenvectors of $PP^\mathsf{T}$ |
| $W$ | Model parameters |
| $\theta$ | Parameters of the proxy distribution |
| $\mathbf{X}$ | Training Data |
| $\mathbf{Y}$ | Training Labels |
| $x$ | Test data sample |
| $y$ | Test label sample |
| $p$ | Probability distribution |
| $q$ | proxy distribution |
| $\mathcal{L}$ | Loss |
| $\mathcal{N}$ | Normal distribution |
| $\mathbb{E}$ | Expectation |
| Cov | Covariance Matrix |
| $[.\ \ .]$ | Column wise Block Concatenation |
| $\lfloor . \rfloor$ | Stop Gradient Function |
| $(.)^\mathsf{T}$ | Transposed Matrix |
| $(.)^a$ | Symbol representing aleatoric uncertainty only |
| $(.)^e$ | Symbol representing epistemic uncertainty only |
| $(.)_W$ | Symbol is a function of the weights |
| $(.)_{w_i}$ | Symbol uses the set of weights $w_i$ |
| $(.)_{i\_}$ | $i^{\text{th}}$ row of a matrix |
| $(.)_{\_i}$ | $i^{\text{th}}$ column of a matrix |

### A.2    LIST OF ACRONYMS

#### ACRONYMS

**AUC**  Area Under the Curve

**CelebA**  CelebFaces Attributes

**D**  diagonal

**DE**  Deep Ensemble

**Flying Chairs**

**GT**  ground truth

**ID**  in-distribution

**KS**  Kolmogorov–Smirnov

**LA**  Laplace Approximation

**LR+D**  low-rank plus diagonal

**LU**  lower-upper

**MAP**  Maximum a posteriori

**MC**  Monte Carlo

**MCD**  Monte Carlo Dropout

**MNIST** Modified National Institute of Standards and Technology database

**NLL** negative log-likelihood

**OOD** out-of-distribution

**SVD** Singular Value Decomposition

**SVI** Stochastic Variational Inference

# B  COMPUTATION, TIME AND SPACE COMPLEXITY

| Type | Parametrization | Captured Corr. | Parametr. Represent. | Memory | Time $x\Sigma^{-1}x^{\mathsf{T}}$ | $|\Sigma|$ | Sampling |
|---|---|---|---|---|---|---|---|
| full Russell & Reale (2021) | correlation | all | $\Sigma$ | $S^2$ | $S^3$ | $S^3$ | $S^3$ |
| full Gundavarapu et al. (2019) | Cholesky | all | $\Sigma$ | $S^2$ | $S^2$ | $S$ | $S^2$ |
| sparse Dorta et al. (2018a;b) | inv. band Cholesky | local | $\Sigma^{-1}$ | $SR$ | $SR$ | $S$ | $SR^2$ |
| sparse Monteiro et al. (2020) | LR+D | global | $\Sigma$ | $SR$ | $SR^2$ | $SR^2$ | $SR$ |
| factorized Kendall & Gal (2017) | diagonal | none | $\Sigma$ | $S$ | $S$ | $S$ | $S$ |

Table 4: This table depicts the computational complexity for calculations using different parametrizations for covariance matrices. We use the sparse LR+D parametrization as the basis for our method. This reduces time and spacial complexity in comparison to the naive or Cholesky decomposition and allows for global correlation in comparison to the sparse inverse band Cholesky parametrization. The type and amount of correlations of different parametrization is different (Captured Corr). Furthermore, the used representation enables for efficient calculation of $\Sigma$ or $\Sigma^{-1}$ (Parametr. Representation) and needs different amount of memory. The time complexity is given for calculation of the mahalanobis distance $x\Sigma^{-1}x^{\mathsf{T}}$, determinant $|\Sigma|$ as well as sampling.

Table 4 give the theoretical time and memory complexities of various covariance parametrizations and calculations. The sparse representations are more efficient in terms of memory and computational complexity. However, they do not provide all degrees of freedom of a covariance matrix and are limited to either local or the most important global correlations.

# C  ADDITIONAL RESULTS

## C.1  QUALITATIVE RESULTS

We provide additional qualitative results for every performed tasks. Figures 6 presents optical flow on Flying Chairs, 7 depicts CelebA inpainting, 8 shows CelebA colorization, and 9 illustrates MNIST inpainting. Figure 10 compares both, eigenvectors in both random orientations as well as the different used Bayesian methods.

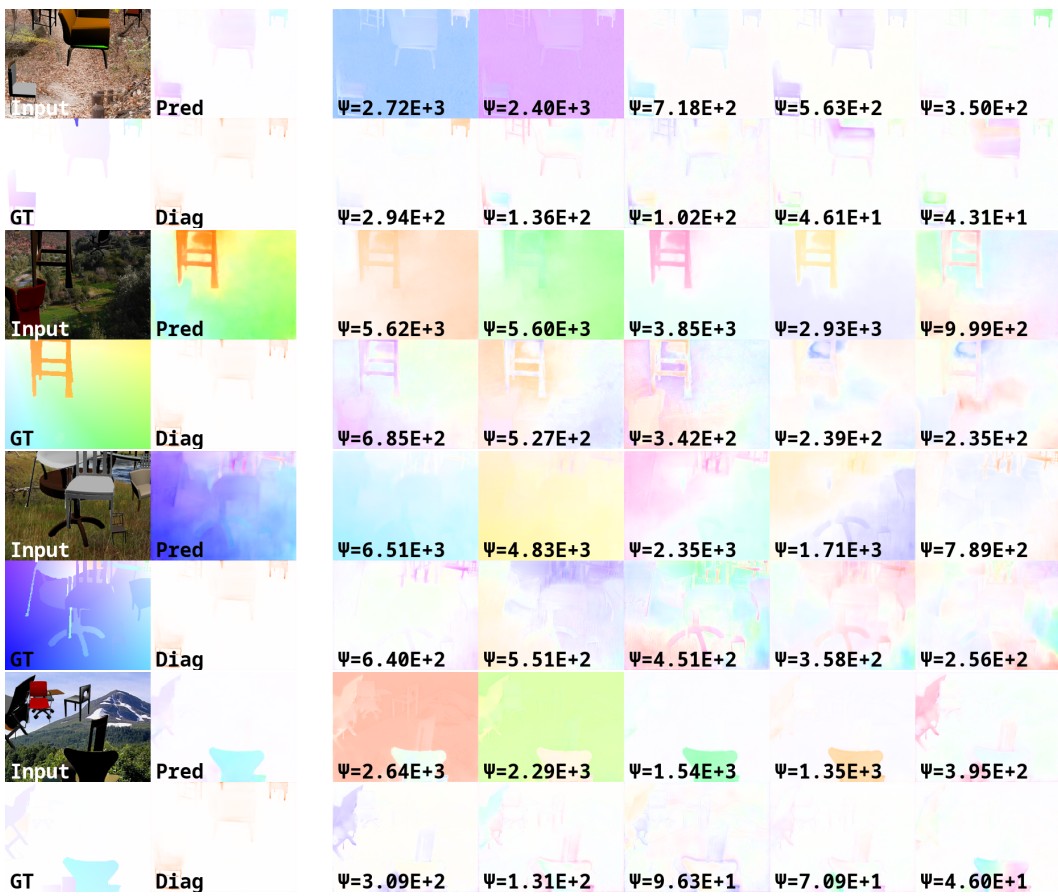

Figure 6: Additional Qualitative Results, Visualizing Flying Chairs' Optical Flow. Random samples from the test sets showing input, prediction, ground truth and parameters of the predictive distribution. The task here is optical flow estimation in the Flying Chairs dataset. The model predicts a mean (Pred), and the parameter $D$ (Diag), as well as a low-rank matrix $P$. In all cases, the predicted joint low-rank matrix $P$ is reduced to the 64 most significant directions (columns) and displayed using the 10 most significant ones in descending order of associated eigenvalues. We can clearly see that the columns focus on uncertainty in certain images areas or colors. Furthermore, the singular values $\Psi$ give a measure of importance of the associated direction. Note that the orientation of the singular vectors is arbitrarily chosen and can be inverted, which results in opposite colors (left) and brightness (right). These eigenvectors are only possible to visualize when modeling covariances and show the direction of maximum variability of the data and helps to understand the underlying factors. Furthermore, we show the upper bound of the angles between the directions of the eigenvectors of $PP^{\mathsf{T}}$ and the eigenvectors of $\Sigma$.

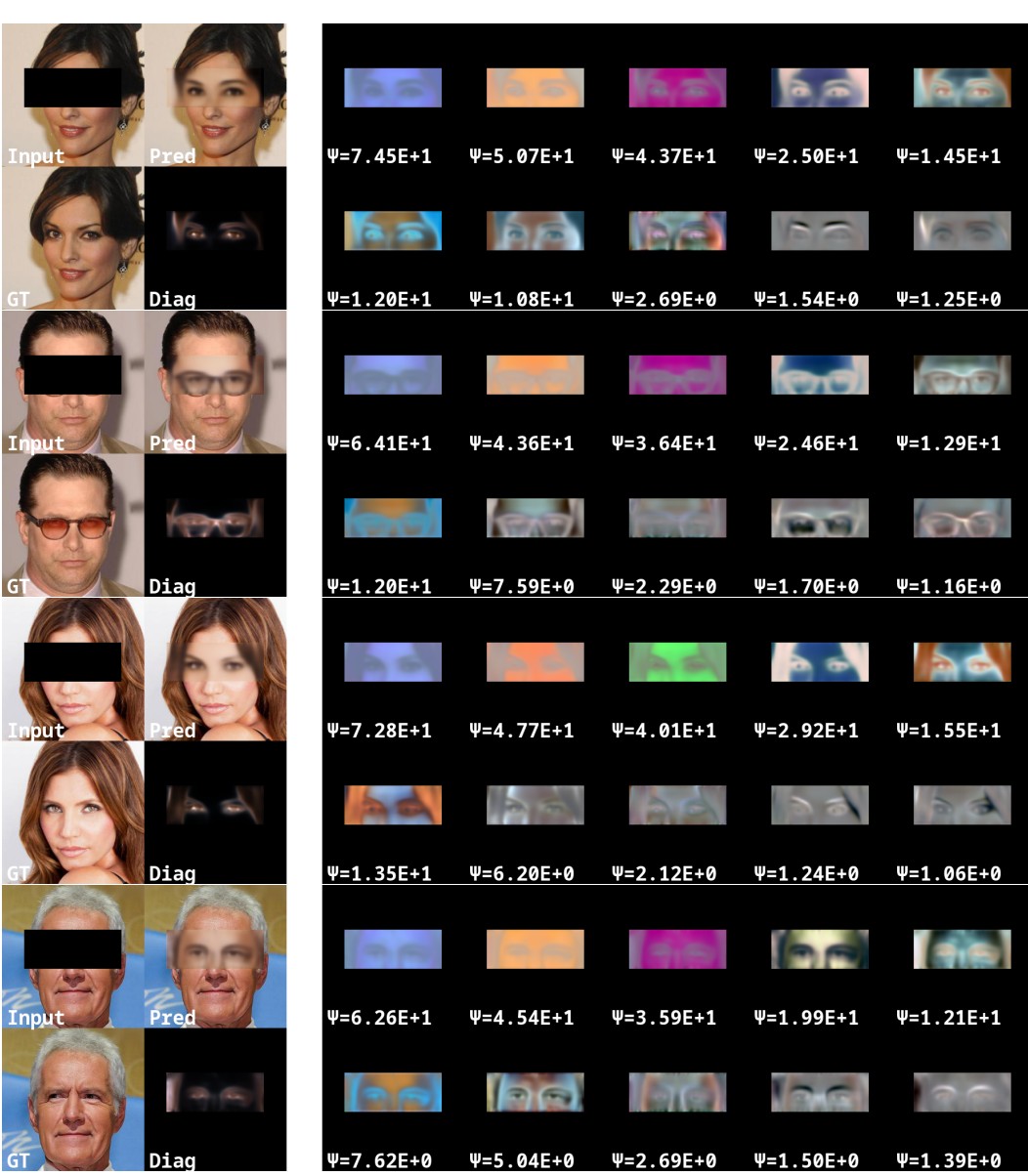

Figure 7: Additional Qualitative Results, Visualizing Inpainting of Eyes of CelebA Faces. Random samples from the test sets showing input, prediction, ground truth and parameters of the predictive distribution. The task here is inpainting of the eyes region of the CelebA faces dataset. The model predicts a mean (Pred), and the parameter $D$ (Diag), as well as a low-rank matrix $P$. In all cases, the predicted joint low-rank matrix $P$ is reduced to the 64 most significant directions (columns) and displayed using the 10 most significant ones in descending order of associated eigenvalues.

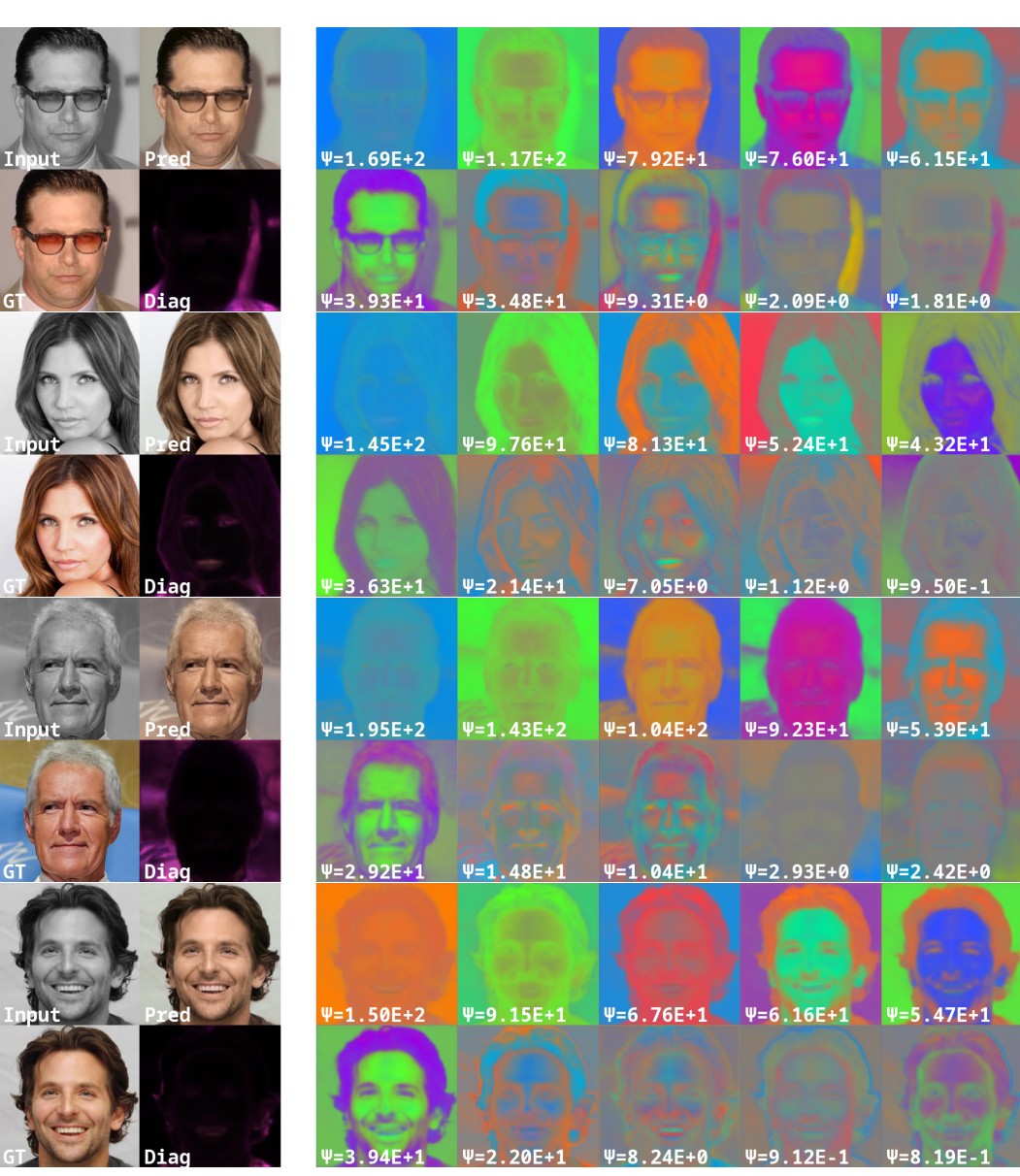

Figure 8: Additional Qualitative Results, Visualizing the Colorization of CelebA Faces. Random samples from the test sets showing input, prediction, ground truth and parameters of the predictive distribution. The task here is colorization of the CelebA faces dataset. The model predicts a mean (Pred), and the parameter $D$ (Diag), as well as a low-rank matrix $P$. In all cases, the predicted joint low-rank matrix $P$ is reduced to the 64 most significant directions (columns) and displayed using the 10 most significant ones in descending order of associated eigenvalues.

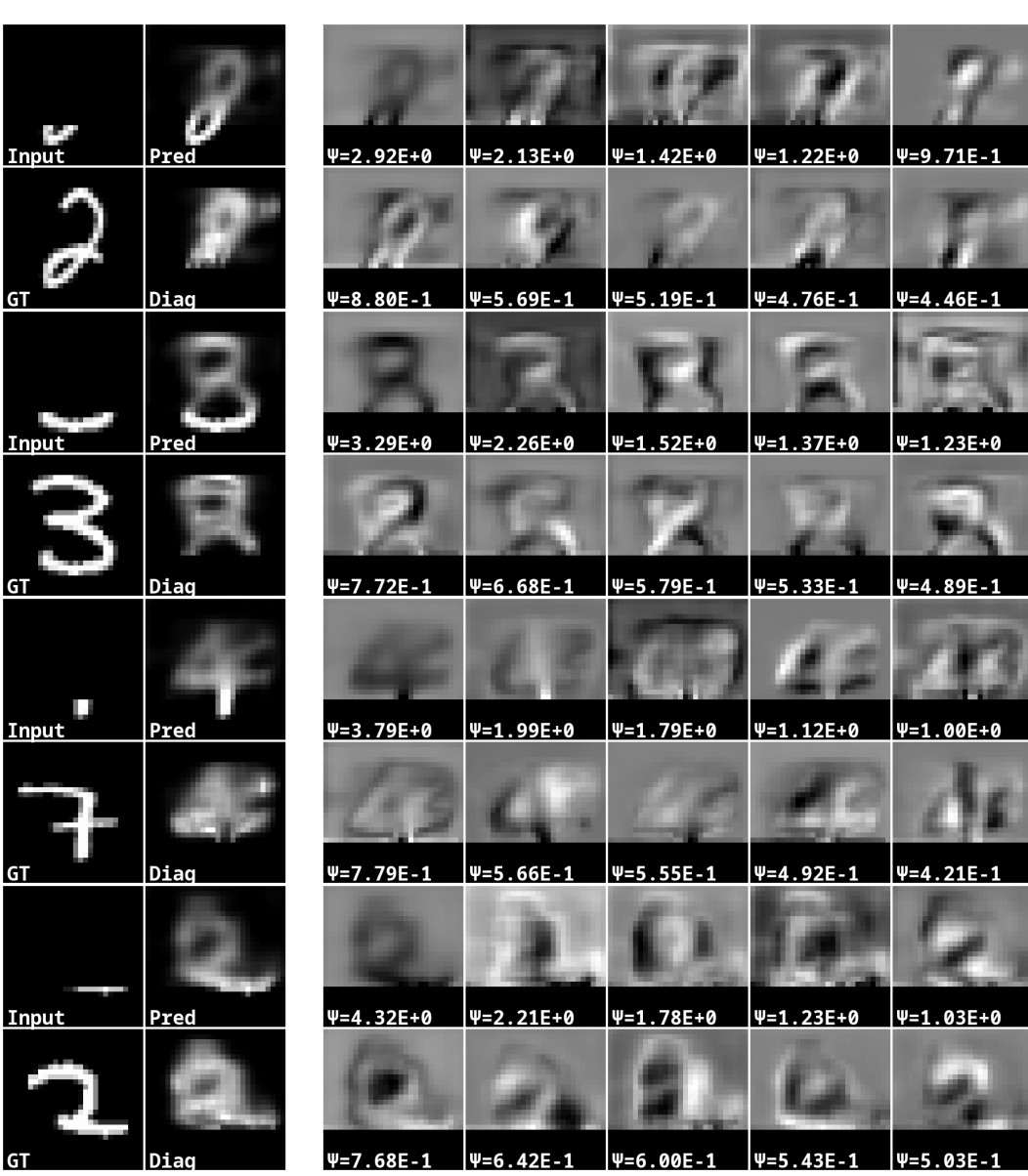

Figure 9: Additional Qualitative Results, Visualizing Inpainting of MNIST Digits. Random samples from the test sets showing input, prediction, ground truth and parameters of the predictive distribution. The task is inpainting MNIST digits. The model predicts a mean (Pred), and the parameter $D$ (Diag), as well as a low-rank matrix $P$. In all cases, the predicted joint low-rank matrix $P$ is reduced to the 64 most significant directions (columns) and displayed using the 10 most significant ones in descending order of associated eigenvalues.

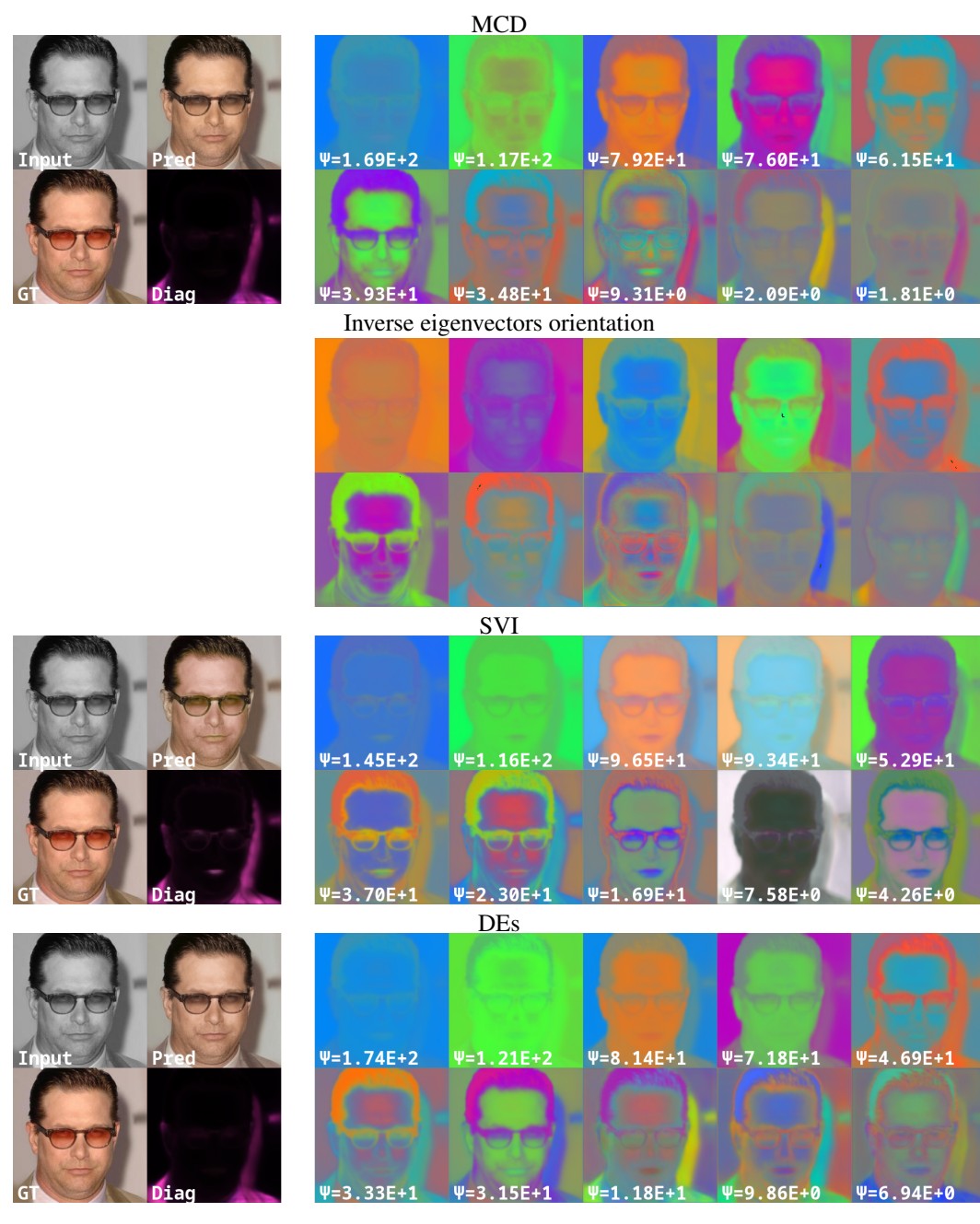

Figure 10: Additional Qualitative Results, comparing Bayesian Methods. A Random sample from the test sets showing input, prediction, ground truth and parameters of the predictive distribution with various Bayesian Methods. For the first method, we also show the eigenvectors with inverse sign. Both signs are mathematically equivalent and one of them is randomly chosen for the visualizations. The task here is colorization of the CelebA faces dataset. The model predicts a mean (Pred), and the parameter $D$ (Diag), as well as a low-rank matrix $P$. In all cases, the predicted joint low-rank matrix $P$ is reduced to the 64 most significant directions (columns) and displayed using the 10 most significant ones in descending order of associated eigenvalues.

## C.2 Quantitative Prediction Errors

Table 5 lists the predicted errors for all bayesian mehtods. We aim for similar predictive errors for all models to get mainly evaluate the quality of the uncertainty using negative log-likelihood (NLL).

| | | | MNIST Inpainting | | CelebA Colorization | | Inpainting | | Flying Chairs Opt. Flow | |
|---|---|---|---|---|---|---|---|---|---|---|
| Epistemic | Param. | $R_W$ | $L_1 \downarrow$ | $L_2 \downarrow$ | $L_1 \downarrow$ | $L_2 \downarrow$ | $L_1 \downarrow$ | $L_2 \downarrow$ | $L_1 \downarrow$ | $L_2 \downarrow$ |
| ✗ | D | 0 | 0.0331 | 0.0475 | 1.9002 | 2.4168 | 2.64 | 5.55 | 0.150 | 0.346 |
| MCD | D | 0 | 0.0335 | 0.0477 | 1.8976 | 2.4159 | 2.58 | 5.62 | 0.148 | 0.325 |
| SVI | D | 0 | 0.0366 | 0.0501 | 1.9074 | 2.4176 | 2.71 | 5.71 | 0.161 | 0.330 |
| DE | D | 0 | 0.0339 | 0.0483 | 1.8977 | 2.4159 | 2.57 | 5.63 | 0.147 | 0.325 |
| ✗ | LR+D | 8 | 0.0352 | 0.0494 | 1.9330 | 2.4244 | 2.59 | 5.52 | 0.176 | 0.345 |
| MCD | LR+D | 4 | 0.0379 | 0.0530 | 1.9116 | 2.4183 | 2.51 | 5.52 | 0.178 | 0.338 |
| MCD | LR+D | 8 | 0.0340 | 0.0481 | 1.9191 | 2.4206 | 2.56 | 5.52 | 0.157 | 0.328 |
| MCD | LR+D | 16 | 0.0338 | 0.0480 | 1.9182 | 2.4202 | 2.61 | 5.57 | 0.155 | 0.328 |
| SVI | LR+D | 8 | 0.0384 | 0.0529 | 1.9494 | 2.4297 | 2.67 | 5.65 | 0.162 | 0.333 |
| DE | LR+D | 8 | 0.0348 | 0.0489 | 1.9219 | 2.4213 | 2.60 | 5.60 | 0.158 | 0.329 |

Table 5: Comparison of reconstruction or prediction errors of all methods. We use the same loss for the prediction between those methods. The last convolutional layer of models with LR+D parametrization has more channels in comparison to models with diagonal parametrization. Furthermore, the uncertainty channels receive gradients from different negative log likelihood functions. Bayesian models (Epistemic) include additional Dropout layer or variational convolutional layer and are evaluated using 64 weight samples. Essentially, the presented study shows our robust, better uncertainty quantification towards the quality of the prediction. This is important to evaluate because the negative-log likelihood is affected by both prediction and uncertainty estimation.

## C.3 Additional Ablation Study

One component of our proposed method is to retain the variance of the removed columns after dimensionality reduction using SVD, see 6. In Table 6 we ablate this design choice. The column $\hat{D}$ indicates whether the diagonal $D$ is updated (✓) after performing SVD according to Equations 6 and 10, or if the original $D$ is retained (✗) as per Equation 8. Our ablation shows that updating the diagonal $D$ appears to slightly improve the average, NLL while also enhancing prediction consistency and reducing test set variability. We show that this is consistent for three different configurations of dimensionality reductions, see Table 6.

Table 7 compares the performance of performing SVD independently on the epistemic and aleatoric low-rank matrices $P^e$ and $P^a$ versus on the combined low-rank matrix $P$. The findings show that

| $R$ | $P^a$ | $P^e$ | $\hat{D}$ | MNIST Inpainting $\times 1$ | CelebA Colorization $\times 1000$ | Inpainting $\times 100$ | Flying Chairs Optical Flow $\times 100$ |
|---|---|---|---|---|---|---|---|
| 64 | SVD(32) | SVD(32) | ✓ | -379± 202 | -239±35 | 71±52 | 849±308 |
| 32 | SVD(16) | SVD(16) | ✓ | -350± 164 | -232±39 | 84±57 | 870±322 |
| 64 | joint SVD(64) | | ✓ | -409± 302 | -242±35 | 68±51 | 844±308 |
| 64 | SVD(32) | SVD(32) | ✗ | -408±2087 | -241±41 | 70±55 | 845±324 |
| 32 | SVD(16) | SVD(16) | ✗ | -368±2985 | -233±52 | 85±62 | 869±348 |
| 64 | joint SVD(64) | | ✗ | -410±1980 | -242±39 | 68±53 | 841±321 |

Table 6: Comparison between adapting the diagonal $D$ after performing the SVD according to Equations 6 and 10 or not. Here $R$ denotes the number of columns in the resulting representation. The columns $P^a$ and $P^e$ denote if and to what degree the dimensionality is reduced after sampling using SVD. The numbers in brackets denote the kept singular vectors, which result in columns $R$. The column $\hat{D}$ indicates whether the diagonal $D$ is updated (✓) after performing SVD according to Equations 6 and 10, or if the original $D$ is retained as per Equation 8, despite the dimensionality reduction of $P$. We show in the first three rows that updating the diagonal $D$ appears to slightly improve the average NLL while also enhancing prediction consistency and reducing test set variability.

| $R$ | $P^a$ | $P^e$ | MNIST Inpainting $\times 1$ | CelebA Colorization $\times 1000$ | Inpainting $\times 100$ | Flying Chairs Optical Flow $\times 100$ |
|---|---|---|---|---|---|---|
| 64 | SVD(32) | SVD(32) | -379± 202 | -239± 35 | 71± 52 | 849±308 |
| 32 | SVD(16) | SVD(16) | -350± 164 | -232± 39 | 84± 57 | 870±322 |
| 16 | SVD( 8) | SVD( 8) | -322± 158 | -222± 41 | 99± 64 | 892±333 |
| 64 | | joint SVD(64) | -409± 302 | -242± 35 | 68± 51 | 844±308 |
| 32 | | joint SVD(32) | -372± 204 | -235± 41 | 82± 57 | 866±322 |
| 16 | | joint SVD(16) | -345± 170 | -228± 45 | 96± 63 | 888±336 |
| 8 | | joint SVD( 8) | -319± 165 | -217± 48 | 113± 71 | 915±349 |

Table 7: Comparison of different approximations of the LR+D-parametrized covariance matrix using varying degrees of dimensionality reduction via SVD. $T$ denotes the number of samples drawn from $q^\star(w)$, and $R$ indicates the number of columns in the resulting representation. The columns $P^e$ and $P^a$ denote on which matrix SVD is performed and the degree of dimensionality reduction. The numbers in brackets represent the retained singular vectors, resulting in columns $R$. If the SVD is performed on the joint matrix $P$ instead of the epistemic and aleatoric submatrices $P^e/P^a$, it is marked as joint. We observe that with the same number of columns $R$, performing SVD on the joint matrix $P$ yields better performance. Conversely, when retaining the same number of columns per SVD, the separate version performs slightly better. In both cases, a higher number of columns is preferable.

| $R^W$ | $R$ | $P$ | MNIST Inpainting $\times 1$ | CelebA Colorization $\times 1000$ | Inpainting $\times 100$ | Flying Chairs Optical Flow $\times 100$ |
|---|---|---|---|---|---|---|
| 4 | 16 | joint SVD(16) | -193± 63 | -198± 39 | 147± 62 | 892±412 |
| 8 | 16 | joint SVD(16) | -345± 170 | -228± 45 | 96± 63 | 888±336 |
| 16 | 16 | joint SVD(16) | -363± 156 | -206± 37 | 224± 136 | 895±322 |
| 4 | 32 | joint SVD(32) | -241± 73 | -207± 32 | 113± 52 | 870±387 |
| 8 | 32 | joint SVD(32) | -372± 204 | -235± 41 | 82± 57 | 866±322 |
| 16 | 32 | joint SVD(32) | -393± 196 | -217± 35 | 162± 122 | 863±310 |
| 8 | 64 | joint SVD(64) | -409± 302 | -242± 35 | 68± 51 | 844±308 |
| 16 | 64 | joint SVD(64) | -423± 254 | -228± 31 | 114± 111 | 833±302 |

Table 8: Comparison between different number of columns used during training the model. While $R^W$ denotes the number of columns produced by the model without sampling., $R$ denotes the number of columns in the resulting representation. The column $P$ show how SVD is used dimensionality is reduced. The number in the brackets denote the kept singular vectors, which result as columns $R$. The results tend to be better with a higher number of learned columns $R^W$. In genreal, increasing $R^W$ can cause increasing training time. However, we also experienced instabilities during training for 3 of those tasks when using $R^W = 32$.

combined SVD yields better results for the same number of columns. However, for a fixed number of singular vectors retained per SVD, the independent approach performs slightly better. Note that combined SVD complicates the post-hoc separation of aleatoric and epistemic contributions.

Furthermore, Table 8 evaluates the impact of the number of columns predicted by the model's output layer. Increasing the number of columns generally benefits the NLL. However, this increases computational complexity and may lead to numerical instabilities during training, as observed in 3 out of 4 tasks failing with $R^W = 32$ columns. Balancing these trade-offs, we chose a rank of 8, which aligns with the choice of Monteiro et al. (2020).

Finally, Table 9 presents an extended ablation of various parameters, non-Bayesian with various Bayesan Models (epis) and both kinds of distribution parametrizations (Param). Therefore, it compares a purely diagonal (D) uncertainty with our LR+D parametrization. The representation took $T$ Bayesian samples, and results in $R$ columns of the low-rank matrix. The number of samples can be reduced for the aleatoric matrix ($P^a$) the epistemic matrix ($P^e$) or both together (in the center of both columns). The columns are either not reduced (-) reduced using Singular Value Decomposition (SVD) with the remaining number of columns in the brackets, or for the aleatoric covariance matrix using the expected weights $\mathbb{E}[W]$. The column $\hat{D}$ indicates whether the diagonal $D$ is updated (✓) after

| Epis | Param | $R^W$ | $T$ | $R$ | $P^a$ | $P^e$ | $\hat{D}$ | MNIST Inpainting ×1 | CelebA Colorization ×1000 | CelebA Inpainting ×100 | Flying Chairs Optical Flow ×100 |
|---|---|---|---|---|---|---|---|---|---|---|---|
| ✗ | D | | 0 + 1 | 0 | - | 0 | - | 2665± 7794 | -146± 111 | 153± 175 | 1059± 496 |
| ✗ | LR+D | 8 | 0 + 1 | 8 | - | 0 | - | 2610±24982 | -216± 78 | 134± 70 | 882± 554 |
| MCD | D | | 64 + 1 | 0 | $\mathbb{E}[W]$ | - | - | -253± 589 | -151± 80 | 128± 152 | 1014± 402 |
| MCD | D | | 64 + 0 | 0 | - | - | - | -292± 345 | -152± 77 | 125± 146 | 1012± 390 |
| MCD | LR+D | 8 | 64 + 1 | 72 | $\mathbb{E}[W]$ | - | - | -155± 4257 | -235± 41 | 73± 62 | 853± 342 |
| MCD | LR+D | 8 | 64 + 0 | 576 | - | - | - | -455± 994 | ** | ** | ** |
| MCD | LR+D | 8 | 32 + 0 | 288 | - | - | - | -440± 1401 | *-250± 24 | 52± 46 | 813± 289 |
| MCD | LR+D | 8 | 16 + 0 | 144 | - | - | - | -411± 2035 | -244± 29 | 64± 51 | 838± 311 |
| MCD | LR+D | 8 | 8 + 0 | 72 | - | - | - | -352± 3095 | -237± 37 | 80± 59 | 868± 341 |
| MCD | LR+D | 8 | 64 + 0 | 64 | SVD(32) | SVD(32) | ✓ | -379± 202 | -238± 29 | 69± 52 | 849± 308 |
| MCD | LR+D | 8 | 64 + 0 | 32 | SVD(16) | SVD(16) | ✓ | -350± 164 | -231± 34 | 83± 58 | 870± 322 |
| MCD | LR+D | 8 | 64 + 0 | 16 | SVD( 8) | SVD( 8) | ✓ | -322± 158 | -222± 36 | 97± 67 | 892± 333 |
| MCD | LR+D | 8 | 64 + 0 | 64 | joint SVD(64) | | ✓ | -409± 302 | -241± 28 | 66± 52 | 844± 308 |
| MCD | LR+D | 8 | 64 + 0 | 64 | joint SVD(64) | | ✗ | -410± 1979 | -242± 30 | 66± 53 | 841± 321 |
| MCD | LR+D | 8 | 64 + 0 | 32 | joint SVD(32) | | ✓ | -372± 204 | -234± 32 | 80± 57 | 866± 322 |
| MCD | LR+D | 8 | 64 + 0 | 32 | joint SVD(32) | | ✗ | -372± 2773 | -236± 36 | 80± 61 | 864± 345 |
| MCD | LR+D | 8 | 64 + 0 | 16 | joint SVD(16) | | ✓ | -345± 170 | -227± 37 | 95± 64 | 888± 336 |
| MCD | LR+D | 8 | 64 + 0 | 16 | joint SVD(16) | | ✗ | -326± 3914 | -229± 47 | 96± 71 | 890± 371 |
| MCD | LR+D | 8 | 64 + 0 | 8 | joint SVD( 8) | | ✓ | -319± 165 | -218± 40 | 111± 74 | 915± 349 |
| MCD | LR+D | 8 | 64 + 0 | 8 | joint SVD( 8) | | ✗ | -273± 5523 | -219± 59 | 116± 86 | 930± 417 |
| MCD | LR+D | 4 | 64 + 1 | 68 | $\mathbb{E}[W]$ | - | - | -376± 1677 | -218± 31 | 77± 55 | 859± 422 |
| MCD | LR+D | 4 | 64 + 0 | 320 | - | - | - | ** | *-230± 21 | 51± 38 | 807± 299 |
| MCD | LR+D | 4 | 32 + 0 | 160 | - | - | - | *-396± 420 | -227± 24 | 62± 42 | 831± 325 |
| MCD | LR+D | 4 | 16 + 0 | 80 | - | - | - | -396± 848 | -222± 27 | 78± 50 | 862± 369 |
| MCD | LR+D | 4 | 8 + 0 | 40 | - | - | - | -382± 1396 | -216± 31 | 98± 60 | 894± 412 |
| MCD | LR+D | 4 | 64 + 0 | 64 | SVD(32) | SVD(32) | ✓ | -245± 73 | -222± 26 | 71± 47 | 849± 351 |
| MCD | LR+D | 4 | 64 + 0 | 32 | SVD(16) | SVD(16) | ✓ | -198± 62 | -217± 28 | 87± 53 | 872± 379 |
| MCD | LR+D | 4 | 64 + 0 | 16 | SVD( 8) | SVD( 8) | ✓ | -161± 55 | -211± 30 | 104± 59 | 895± 405 |
| MCD | LR+D | 4 | 64 + 0 | 32 | joint SVD(32) | | ✓ | -241± 73 | -218± 28 | 84± 52 | 870± 387 |
| MCD | LR+D | 4 | 64 + 0 | 32 | joint SVD(32) | | ✗ | -400± 1092 | -219± 29 | 84± 54 | 870± 406 |
| MCD | LR+D | 4 | 64 + 0 | 16 | joint SVD(16) | | ✓ | -193± 63 | -213± 30 | 101± 59 | 892± 412 |
| MCD | LR+D | 4 | 64 + 0 | 16 | joint SVD(16) | | ✗ | -399± 1372 | -214± 32 | 102± 62 | 896± 454 |
| MCD | LR+D | 4 | 64 + 0 | 8 | joint SVD( 8) | | ✓ | -157± 56 | -206± 33 | 117± 65 | 914± 432 |
| MCD | LR+D | 4 | 64 + 0 | 8 | joint SVD( 8) | | ✗ | -395± 1635 | -208± 37 | 120± 72 | 924± 492 |
| MCD | LR+D | 16 | 64 + 1 | 80 | $\mathbb{E}[W]$ | - | - | -208± 3830 | -240± 80 | 56± 73 | 820± 365 |
| MCD | LR+D | 16 | 64 + 0 | 1088 | - | - | - | -483± 725 | ** | ** | ** |
| MCD | LR+D | 16 | 32 + 0 | 544 | - | - | - | -473± 956 | ** | * 27± 47 | * 760± 283 |
| MCD | LR+D | 16 | 16 + 0 | 272 | - | - | - | -446± 1417 | *-259± 33 | 36± 52 | 791± 303 |
| MCD | LR+D | 16 | 8 + 0 | 136 | - | - | - | -403± 2126 | -251± 45 | 50± 60 | 819± 330 |
| MCD | LR+D | 16 | 64 + 0 | 64 | SVD(32) | SVD(32) | ✓ | -399± 190 | -244± 37 | 57± 60 | 843± 299 |
| MCD | LR+D | 16 | 64 + 0 | 32 | SVD(16) | SVD(16) | ✓ | -368± 155 | -234± 42 | 71± 67 | 872± 306 |
| MCD | LR+D | 16 | 64 + 0 | 64 | joint SVD(64) | | ✓ | -423± 254 | -247± 35 | 54± 60 | 833± 302 |
| MCD | LR+D | 16 | 64 + 0 | 64 | joint SVD(64) | | ✗ | -424± 1930 | -248± 45 | 54± 66 | 817± 332 |
| MCD | LR+D | 16 | 64 + 0 | 32 | joint SVD(32) | | ✓ | -393± 196 | -239± 40 | 71± 68 | 863± 310 |
| MCD | LR+D | 16 | 64 + 0 | 32 | joint SVD(32) | | ✗ | -375± 2892 | -238± 61 | 72± 81 | 843± 360 |
| MCD | LR+D | 16 | 64 + 0 | 16 | joint SVD(16) | | ✓ | -363± 156 | -229± 46 | 91± 74 | 895± 322 |
| MCD | LR+D | 16 | 64 + 0 | 16 | joint SVD(16) | | ✗ | -339± 3547 | -225± 101 | 97± 96 | 877± 410 |
| SVI | D | | 64 + 1 | 0 | $\mathbb{E}[W]$ | - | - | -263± 311 | -157± 35 | 126± 81 | 1045± 386 |
| SVI | D | | 64 + 0 | 0 | - | - | - | -268± 297 | -157± 35 | 125± 79 | 1043± 381 |
| SVI | LR+D | 8 | 64 + 1 | 72 | $\mathbb{E}[W]$ | - | - | -327± 3232 | -225± 38 | 116± 35 | 760± 524 |
| SVI | LR+D | 8 | 64 + 0 | 576 | - | - | - | -381± 2638 | ** | 100± 29 | ** |
| SVI | LR+D | 8 | 32 + 0 | 288 | - | - | - | -371± 2810 | ** | 104± 30 | * 723± 399 |
| SVI | LR+D | 8 | 16 + 0 | 144 | - | - | - | -358± 3056 | *-233± 28 | 109± 32 | 733± 463 |
| SVI | LR+D | 8 | 8 + 0 | 72 | - | - | - | -342± 3340 | -226± 35 | 118± 34 | 770± 508 |
| SVI | LR+D | 8 | 64 + 0 | 64 | SVD(32) | SVD(32) | ✓ | -396± 2374 | -226± 32 | 111± 32 | 754± 479 |
| SVI | LR+D | 8 | 64 + 0 | 32 | SVD(16) | SVD(16) | ✓ | -404± 2121 | -217± 33 | 122± 36 | 787± 501 |
| SVI | LR+D | 8 | 64 + 0 | 16 | SVD( 8) | SVD( 8) | ✓ | -399± 1603 | -194± 23 | 135± 38 | 826± 509 |
| SVI | LR+D | 8 | 64 + 0 | 64 | joint SVD(64) | | ✓ | -383± 2591 | -229± 31 | 109± 31 | 748± 467 |
| SVI | LR+D | 8 | 64 + 0 | 64 | joint SVD(64) | | ✗ | -348± 3251 | -229± 33 | 108± 31 | 744± 505 |
| SVI | LR+D | 8 | 64 + 0 | 32 | joint SVD(32) | | ✓ | -396± 2355 | -222± 35 | 118± 34 | 780± 506 |
| SVI | LR+D | 8 | 64 + 0 | 32 | joint SVD(32) | | ✗ | -333± 3514 | -223± 39 | 118± 34 | 778± 567 |
| SVI | LR+D | 8 | 64 + 0 | 16 | joint SVD(16) | | ✓ | -404± 2051 | -212± 37 | 132± 39 | 816± 529 |
| SVI | LR+D | 8 | 64 + 0 | 16 | joint SVD(16) | | ✗ | -319± 3777 | -214± 48 | 132± 40 | 817± 609 |
| SVI | LR+D | 8 | 64 + 0 | 8 | joint SVD( 8) | | ✓ | -396± 1607 | -191± 25 | 147± 43 | 860± 537 |
| SVI | LR+D | 8 | 64 + 0 | 8 | joint SVD( 8) | | ✗ | -303± 4061 | -205± 57 | 146± 46 | 863± 650 |
| DE | D | | 64 + 0 | 0 | - | - | - | -308± 236 | -158± 58 | 93± 101 | 965± 291 |
| DE | LR+D | 8 | 64 + 0 | 576 | - | - | - | -483± 337 | ** | * 43± 37 | ** |
| DE | LR+D | 8 | 32 + 0 | 288 | - | - | - | -487± 604 | *-252± 23 | 47± 41 | 805± 274 |
| DE | LR+D | 8 | 16 + 0 | 144 | - | - | - | -465± 1155 | -245± 27 | 60± 48 | 826± 297 |
| DE | LR+D | 8 | 8 + 0 | 72 | - | - | - | -339± 3213 | -237± 36 | 80± 60 | 868± 344 |
| DE | LR+D | 8 | 64 + 0 | 64 | SVD(32) | SVD(32) | ✓ | -358± 93 | -233± 25 | 64± 43 | 845± 270 |
| DE | LR+D | 8 | 64 + 0 | 32 | SVD(16) | SVD(16) | ✓ | -321± 82 | -212± 24 | 86± 39 | 873± 269 |
| DE | LR+D | 8 | 64 + 0 | 16 | SVD( 8) | SVD( 8) | ✓ | -280± 72 | -184± 22 | 112± 34 | 911± 264 |
| DE | LR+D | 8 | 64 + 0 | 64 | joint SVD(64) | | ✓ | -394± 118 | -240± 25 | 61± 44 | 837± 272 |
| DE | LR+D | 8 | 64 + 0 | 64 | joint SVD(64) | | ✗ | -490± 703 | -243± 28 | 60± 45 | 827± 281 |
| DE | LR+D | 8 | 64 + 0 | 32 | joint SVD(32) | | ✓ | -351± 93 | -229± 27 | 79± 48 | 862± 276 |
| DE | LR+D | 8 | 64 + 0 | 32 | joint SVD(32) | | ✗ | -491± 908 | -235± 34 | 76± 54 | 846± 295 |
| DE | LR+D | 8 | 64 + 0 | 16 | joint SVD(16) | | ✓ | -313± 82 | -210± 25 | 100± 42 | 892± 275 |
| DE | LR+D | 8 | 64 + 0 | 16 | joint SVD(16) | | ✗ | -487± 1093 | -230± 39 | 90± 61 | 862± 309 |
| DE | LR+D | 8 | 64 + 0 | 8 | joint SVD( 8) | | ✓ | -273± 73 | -183± 22 | 123± 35 | 927± 269 |
| DE | LR+D | 8 | 64 + 0 | 8 | joint SVD( 8) | | ✗ | -477± 1229 | -217± 45 | 103± 67 | 878± 326 |

Table 9: Extended Ablation. We compare non-Bayesian networks with aleatoric uncertainty only and various Bayesian networks with both kind of uncertainties and various hyperparameters.

performing SVD according to Equations 6 and 10, or if the original $D$ is retained as per Equation 8, despite the dimensionality reduction of $P$. Drawing many samples without any rank compression can make the approach numerically unstable. Here, $\star$ marks ($\star\star$ replaces) values, where less (more) than 2% of the test set results run into numerical errors. To alleviate this, we reduce the dimensionality of the representation and remove the eigenvectors associated with smaller singular values from the low-rank matrix.

## D DERIVATIONS IN DETAIL

### D.1 EXPLOITING LR+D FOR EFFICIENT COMPUTATION OF MATRIX DETERMINANT AND INVERSE

Both the likelihood function $p(y|x, W) = \mathcal{N}\left(\mu_W^a\left(x\right), \Sigma_W^a\left(x\right)\right)$ as well as the approximate posterior predictive distribution $p(y|x, X, Y) \approx \mathcal{N}\left(\mu\left(x\right), \Sigma\left(x\right)\right)$ are multivariate normal distributions parametrized by covariance matrices $\Sigma_W^a$ and $\Sigma$, respectively, where in the following, we only consider $\Sigma$ for clarity. Denoting by $S$ the output dimension, the normal distribution is then defined as

$$\mathcal{N}\left(\mu\left(x\right), \Sigma\left(x\right)\right) = \frac{1}{\sqrt{|\Sigma(x)|(2\pi)^S}} \exp\left(-\frac{1}{2}(\mu(x) - y)^\intercal \Sigma^{-1}(x)(\mu(x) - y)\right) \qquad (11)$$

which requires computation of the covariance matrix' determinant $|\Sigma|$ and inverse $\Sigma^{-1}$ for sampling and evaluation of the log likelihood. For full covariance matrices $\Sigma \in \mathbb{R}^{S \times S}$ with large $S$, these are very expensive, if not impossible, to compute directly. Instead, we exploit our LR+D representation for efficient computation of the matrix determinant and inverse.

We compute the determinant as

$$|\Sigma| = |D + PP^\intercal| \qquad (12)$$
$$= |I_R + P^\intercal D^{-1} P||D| \qquad (13)$$
$$= |C||D| \qquad (14)$$

where we first substituted $\Sigma$ with its LR+D representation and subsequently applied the matrix determinant lemma. With $D \in \mathbb{R}^{S \times S}$, $P \in \mathbb{R}^{S \times R}$ and $I_R \in \mathbb{R}^R$, the so-called capacitance $C = I_R + P^\intercal D^{-1} P$ is an $R \times R$ matrix. Since $R \ll S$, the determinant of the capacitance matrix is very cheap to compute.

To compute the inverse, we use the Woodbury matrix identity, again by exploiting the LR+D representation.

$$\Sigma^{-1} = (D + PP^\intercal)^{-1} \qquad (15)$$
$$= D^{-1} - D^{-1}P(I_R + P^\intercal D^{-1}P)^{-1}P^\intercal D^{-1} \qquad (16)$$
$$= D^{-1} - D^{-1}PC^{-1}P^\intercal D^{-1} \qquad (17)$$

As before, the capacitance matrix $C \in \mathbb{R}^{R \times R}$ is very small and thus its inverse easy to compute.

### D.2 FULL DERIVATION OF SVD

We apply dimensionality reduction using SVD on our tall $P$ matrices. This involves decomposing into three separate matrices: $U$, $\Psi$, and $V^\intercal$. The $U$ matrix represents an arbitrary not further used rotation, $\Psi$ is a diagonal matrix containing the singular values, and $V^\intercal$ contains the columns of the transformed matrix.

By selecting the top $R$ singular values and corresponding vectors, we can approximate the original matrix. This approximation is achieved by truncating the matrices $U$ and $V^\intercal$ to retain only the top $R$ singular values and vectors. This reduces the dimensionality of the data while preserving its essential structure.

The reduced dimensionality representation, denoted as $\hat{P}$, is computed by taking the product of the truncated matrices $V$ and $\Psi$. Additionally, a diagonal matrix $\hat{D}$ captures the by the dimensionality reduction removed variance of $PP^\intercal$, with each element representing the contribution of the omitted

singular values to the overall uncertainty. We use $\hat{D}$ to update our diagonal for the final LR+D representation.

$$P^\intercal = U\Psi V^\intercal \tag{18}$$

$$PP^\intercal = (U\Psi V^\intercal)^\intercal(U\Psi V^\intercal) \tag{19}$$

$$= V\Psi U^\intercal U\Psi V^\intercal \tag{20}$$

$$= V\Psi\Psi V^\intercal \tag{21}$$

$$PP^\intercal = \hat{D} + \hat{P}\hat{P}^\intercal \tag{22}$$

$$\hat{P} = \begin{bmatrix} V_{R-\hat{R}} \cdot \Psi_{R-\hat{R},R-\hat{R}} & \cdots & V_R \cdot \Psi_{R,R} \end{bmatrix} \tag{23}$$

$$\hat{D}_{ii} = \sum_{j=1}^{R-\hat{R}-1} V_{ij}^2 \cdot \Psi_{j,j}^2 \tag{24}$$

### D.3 Loss Definition

For regression problems we intend to maximize the data likelihood $p(\boldsymbol{Y}|\boldsymbol{X}, w) = \prod_i p(y_i|x_i, w)$, where we assumed all dataset samples to be i.i.d. Equivalently, we can minimize the negative log likelihood $p(\boldsymbol{Y}|\boldsymbol{X}, w) = \sum_i -\log p(y_i|x_i, w)$. We further assume the network predictions to be distributed around the true value $y$ following a Gaussian distribution with mean $\mu_w(x)$ and covariance $\Sigma_w(x)$.

The loss function for a single training sample can then be defined as

$$\mathcal{L} = -\frac{1}{S}\log p(y|x, w)$$

$$= -\frac{1}{S}\log\left(\frac{1}{\sqrt{|\Sigma_w|(2\pi)^S}}\exp\left(-\frac{1}{2}(\mu_w - y)^\intercal\Sigma_w^{-1}(\mu_w - y)\right)\right)$$

$$= \frac{1}{S}\left(\log\sqrt{|\Sigma_w|(2\pi)^S} + \frac{1}{2}(\mu_w - y)^\intercal\Sigma_w^{-1}(\mu_w - y)\right)$$

$$= \frac{1}{S}\left(\frac{1}{2}\log|\Sigma_w| + \frac{S}{2}\log(2\pi) + \frac{1}{2}(\mu_w - y)^\intercal\Sigma_w^{-1}(\mu_w - y)\right)$$

where we normalized by the output dimensionality $S$.

Dropping constant terms, we are left with:

$$\mathcal{L} = \frac{1}{2S}\log|\Sigma_w| + \frac{1}{2S}(\mu_w - y)^\intercal\Sigma_w^{-1}(\mu_w - y)$$

We can see that evaluating $\mathcal{L}$ involves computing the determinant and inverse of the covariance matrix. To achieve this, we exploit our LR+D representation as described in the previous section D.1

### D.4 Full derivation of mean vector and covariance matrix

The expectation of the posterior predictive distribution is given by:

$$\mathbb{E}[y|x, \boldsymbol{X}, \boldsymbol{Y}] = \mathbb{E}_{p(W|Y,X)}\left[\mathbb{E}\left[y|x, W\right]\right] \tag{25}$$

$$\approx \mathbb{E}_{q_\theta^*(W)}\left[\mathbb{E}\left[y|x, W\right]\right] \tag{26}$$

$$= \mathbb{E}_{q_\theta^*(W)}\left[\mu_W^a(x)\right] \tag{27}$$

$$\approx \frac{1}{T}\sum_i^T \mu_{w_i}^a(x) \quad w_i \sim q_\theta^*(W) \tag{28}$$

$$= \mu(x) \tag{29}$$

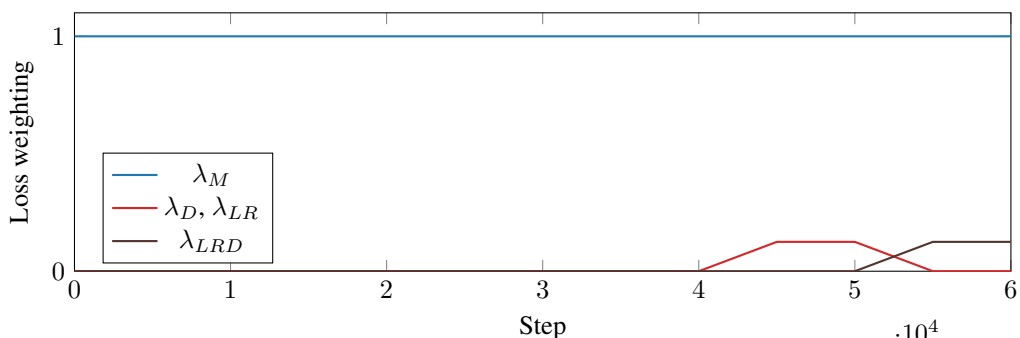

Figure 11: Loss Factors Variation during Training.

The covariance of the posterior predictive distribution is given by:

$$\text{Cov}\left[y|\,x,\boldsymbol{X},\boldsymbol{Y}\right] = \text{Cov}_{p(W|Y,X)}\left[\mathbb{E}_{p(W|Y,X)}\left[y|x,W\right]\right] + \mathbb{E}_{p(W|Y,X)}\left[\text{Cov}\left[y|x,W\right]\right] \tag{30}$$

$$\approx \text{Cov}_{q_\theta^*(W)}\left[\mathbb{E}_{q_\theta^*(W)}\left[y|x,W\right]\right] \quad + \mathbb{E}_{q_\theta^*(W)}\left[\text{Cov}\left[y|x,W\right]\right] \tag{31}$$

$$= \underbrace{\text{Cov}_{q_\theta^*(W)}\left[\mu_W^a(x)\right]}_{epistemic} \quad + \underbrace{\mathbb{E}_{q_\theta^*(W)}\left[\Sigma_W^a(x)\right]}_{aleatoric} \tag{32}$$

$$\approx \Sigma^e(x) \quad + \Sigma^a(x) \tag{33}$$

$$= \Sigma(x) \tag{34}$$

In above transformations of expectation and variance, we applied the law of total expectation or variance, respectively, and subsequently approximated them using the proxy distribution $q_\theta^\star(W)$. The expectation over $y$, denoted by $\mathbb{E}\left[y|x,W\right]$, is given by the mean of the predicted normal distribution $\mu_W^a(x)$, whereas the covariance over $y$, denoted as $\text{Cov}\left[y|x,W\right]$, is given by the covariance matrix of the predicted normal distribution. Finally, the expectation – and in some suggested solutions also the covariance – over the proxy distribution $q_\theta^\star(W)$ is approximated using Monte Carlo integration.

# E  TRAINING SETUP

To stabilize, training, we train our model outputs separately with different loss terms and change the weight of the loss terms over time. We reparametrize the diagonal $D(x) = \text{diag}\left(\exp(s) + 10^{-4}\right)$ to ensure that the diagonal has positive entries and at least a standard deviation of 0.01 in the normalized image domains. Therefore, we combine four losses using the stop gradient operator $\lfloor.\rfloor$ with factors which change over time. Figure 11 shows the gradual increase and decrease losses factors over time to finally train the joint distribution. The whole training lasts 60000 steps where the number of steps per epoch is 1511 for CelebA, 1143 for Flying Chairs, and 195 for MNIST.

$$\mathcal{L} = \lambda_M\,\mathcal{L}_M + \lambda_D\,\mathcal{L}_D + \lambda_{LR}\,\mathcal{L}_{LR} + \lambda_{LRD}\,\mathcal{L}_{LRD} \tag{35}$$

$$\mathcal{L}_M = \log\mathcal{N}\left(\mu\left(x\right),I\right) \tag{36}$$

$$\mathcal{L}_D = \log\mathcal{N}\left(\lfloor\mu\left(x\right)\rfloor,D\left(x\right)\right) \tag{37}$$

$$\mathcal{L}_L = \log\mathcal{N}\left(\lfloor\mu\left(x\right)\rfloor,I + P\left(x\right)P^{\mathsf{T}}\left(x\right)\right) \tag{38}$$

$$\mathcal{L}_{LRD} = \log\mathcal{N}\left(\lfloor\mu\left(x\right)\rfloor,D\left(x\right) + P\left(x\right)P^{\mathsf{T}}\left(x\right)\right) \tag{39}$$

We run all experiments on a single Quattro RTX8000 NVIDIA GPU with 48GB RAM.

