# OpenReview forum: "Structured Joint Aleatoric and Epistemic Uncertainty for High Dimensional Output Spaces"
_ICLR.cc/2025/Conference — ICLR 2025 Conference Withdrawn Submission_

### Official Review · Reviewer_YQBN · 2024-11-01

**Soundness:** 3
**Presentation:** 3
**Contribution:** 3
**Rating:** 3
**Confidence:** 5

**Summary:**

This paper is about aleatoric and epistemic uncertainty estimation in high dimensional output spaces, where the problem is often the dimensionality and representation of covariance matrices, in combination with joint estimation of aleatoric and epistemic uncertainty, which have slightly different estimation processes.

The authors propose to use the LR+D (low rank plus diagonal) formulation for the covariance matrices for aleatoric and epistemic uncertainty in a sparse way, and be able to disentangle aleatoric and epistemic uncertainties using the law of total variance.

Contributions are the use of the LR+D formulation for aleatoric and epistemic uncertainty estimation for sparse representation of covariance matrices within a bayesian neural network framework and modeling correlations between output values, and better results in terms of negative log-likelihood for high dimensional regression tasks (inpainting, colorization, and optical flow estimation), and out of distribution detection on the MNIST inpainting example.

**Strengths:**

- The paper is well written and easy to understand.
- Overall I believe the idea makes sense, to make a trade-off or intermediary point between diagonal covariances and full covariance matrices, for which the LR+D method is appropriate, and in particular I believe applying this formulation to separate estimates of aleatoric and epistemic uncertainty is novel and very interesting.
- The proposed method requires less memory than storing full covariance matrices, and I believe the formulation for making approximations (using naive  and SVD representations) looks correct.
- Results show that the proposed method using LR+D better fits the data overall, with different approximations made in the covariance matrix, for the tasks of MNIST inpainting, CelebA colorization and inpainting, and optical flow estimation on Flying Chairs dataset. There is a good selection of dataset that represent high output dimension regression problems.
- There are results on out of distribution detection, showing that the proposed LR+D approximation is better for out of distribution detection on MNIST inpainting by leaving one class out.
- The baselines cover monte carlo dropout, variational inference and deep ensembles, which is a good selection of commonly used methods for epistemic uncertainty estimation.

**Weaknesses:**

- I believe the evaluation of this paper is incorrect, reading the paper shows formulations for aleatoric and epistemic uncertainty, but these outputs are not evaluated separately, only jointly, and I was expecting that since aleatoric and epistemic covariances are explicitly modeled separately (there are whole sections for each of them), then I would expect that different methodologies be used to evaluate aleatoric and epistemic uncertainty estimates. The evaluation only covers the joint uncertainty (aleatoric + epistemic), and its not clear what is being evaluated. Why have formulations for both kind of uncertainties and then not evaluate them separately?
- The authors use the negative log-likelihood without actually describing the equations for the NLL, but more importantly the interpretation of the NLL is not correct, in Sec 3.2 (Page 6) the authors state: "To evaluate performance, we use the negative log-likelihood, which measures how well a model predicts the observed data." This in my opinion is not completely correct, as the NLL measures how well predictions fit the data but considering uncertainty, it is only about predictions for observed data, the NLL considers uncertainty and this might not be what the authors were expecting. This connects to my previous comment about its unclear what the evaluation methodology is actually evaluating.
- Another big issue with the evaluation is that most results compare models with and without epistemic uncertainty methods applied to them, but this comparison makes no sense as the joint uncertainty is being evaluated, so again it is not clear what this should be measuring, specially since the NLL measures both squared distance between prediction and ground truth, and the output prediction uncertainty together.
- The out of distribution results were a good opportunity to evaluate epistemic uncertainty, as epistemic uncertainty should only be effective for out of distribution detection, and additionally NLL results could be evaluated separately for aleatoric and epistemic uncertainties, and I recommend that for epistemic uncertainty, varying the size of the training set for epistemic uncertainty, and having some kind of ground truth (label noise) for aleatoric uncertainty. Even plotting the aleatoric and epistemic covariances diagonals separately as qualitative results would be a great improvement (just like Kendall and Gal do in their paper).

One minor detail that I do not consider a weakness, is one missing reference to the paper "A deeper look into aleatoric and epistemic uncertainty disentanglement" at CVPR 2022 workshops, which also deals with aleatoric and epistemic uncertainty using the Kendall and Gal formulation but combining with multiple epistemic uncertainty methods.

**Questions:**

- The evaluation focuses on the joint uncertainty, why not evaluate aleatoric and epistemic uncertainties separately?
- What exactly is being evaluated by comparing methods with and without epistemic uncertainty estimation? For example a method without epistemic uncertainty vs an ensemble.

---

### Official Review · Reviewer_hMqf · 2024-11-02

**Soundness:** 3
**Presentation:** 3
**Contribution:** 3
**Rating:** 5
**Confidence:** 5

**Summary:**

The current manuscript introduces a novel way to combine aleatoric (daza uncertainty) and epistemic (model uncertainty) in a joint representation, capturing important interdependencies between output dimensions. The proposed method uses a low-rank plus diagonal (LR+D) covariance matrix to approximate this joint uncertainty, balancing computational efficiency with robust uncertainty estimation. Traditional methods often treat uncertainties independently or assume uncorrelated outputs, which the authors explain can lead to incomplete representations in complex tasks. The LR+D approach is computationally advantageous, reducing memory and processing demands compared to full covariance representations while preserving key correlations. The model has been evaluated using MNIST, CelebA, and Flying Chairs datasets (inpainting, colorization, and optical flow estimation) for IND and OOD detection.

**Strengths:**

* The paper uses a low-rank plus diagonal (LR+D) covariance matrix which seems an efficient method for modeling joint aleatoric and epistemic uncertainties in high-dimensional output spaces, and results in reducing memory and computational requirements compared to full covariance matrices.

* The proposed approach captures correlations between output dimensions, leading to more accurate uncertainty estimation!

* The method has been evaluated and tested on three regression tasks and three dataset and showed improvement.

**Weaknesses:**

* W1: the model can face numerical stability issues, especially when a high number of columns are retained in the low-rank matrix, which may lead to challenges in scaling to very large output spaces or complex tasks without further computational adjustments.

* W2:  the authors presume that a low-rank approximation sufficiently captures the most significant correlations in high-dimensional output spaces. However, this might be inadequate for certain tasks where complex, non-linear dependencies exist across outputs, possibly leading to incomplete uncertainty estimation.

* W3: the experimental settings and achieved results are not sufficient to support the superiority of the method. for example, the authors performed experiments using U-Net architecture and the datasets are small! Moreover, the paper’s approach is compatible with various Bayesian methods, it primarily evaluates Monte Carlo Dropout (MCD), Stochastic Variational inference, and deep ensembles. Other Bayesian methods could potentially enhance uncertainty estimation, and exploring these might provide deeper insights.

**Questions:**

* Q1) Since the Low-rank plus diagonal (LR+D) representation requires empirical tuning of parameters such as the rank of the low-rank matrix and the number of samples for accurate uncertainty representation. Could you please provide a sensitivity analysis showing how *different ranks* and *sample sizes* affect performance and computational requirements?

* Q2) While the LR+D representation reduces the cost compared to full covariance matrices, training the model on high-dimensional tasks with Bayesian frameworks (e.g., requiring Monte Carlo sampling) still demands significant computation, which may limit the method's practical feasibility on large-scale real-world datasets. It would be interesting if the authors could provide more details on computational complexity analyses or empirical runtime comparisons with existing methods on larger datasets.

* Q3) I would encourage authors to show the impact of their method on large-scale datasets and transformer-based architecture, and Please consider more recent approaches for comparison such as [1,2] .

Ref: [1] Mukhoti, Jishnu, et al. "Deep deterministic uncertainty: A new simple baseline." Proceedings of the IEEE/CVF Conference on Computer Vision and Pattern Recognition. 2023.
Ref [2] Daxberger, Erik, et al. "Laplace redux-effortless bayesian deep learning." Advances in Neural Information Processing Systems 34 (2021): 20089-20103.

---

### Official Review · Reviewer_gr6S · 2024-11-04

**Soundness:** 3
**Presentation:** 3
**Contribution:** 2
**Rating:** 5
**Confidence:** 3

**Summary:**

The authors propose a low rank plus diagonal method for capturing joint aleatoric and epistemic uncertainty in high dimensional output spaces.

**Strengths:**

- The method is well derived, and appears to show performance improvements over simple Bayesian baselines.
- As measured by differntial entropy, the method appears to show a better separation of ID and OOD data.

**Weaknesses:**

- If my understanding is correct, the authors promote their method as a way to model both aleatoric and epistemic uncertainty, however, there is no study which disambiguates the two types of uncertainty available in the output prediction.
  - Building on the point above, if equation 2 is completely sum-decomposable, then it might not be possible to disambiguate the two types of uncertainty in the output, which is hard to reconcile with the authors promotion of modeling both epistemic and aleatoric uncertainty. How is it possible in practice to distinguish between aleatoric and epistemic uncertainty?

- Why is the SVD necessary? it seems that $R^W$ is a hyperparameter which can be set as low as desired, so it seems that the current method sets it high, and then truncates  the dimensionality with an SVD. Would it not be better to just set $R^W$ to a lower value in the beginning and save the cost of the SVD decomposition?

- What are the numerical issues which are mentioned in the end? Specifically what happens which causes an issue?

- Can you do a study where you look at the condition numbers of the predicted covariance matrices?

## Overall

Overall, I think the work provides some nice experiments with a nice derivation, however I fail to see the utility of the added novelty over the aleatoric and epistemic uncertainty if those added uncertainties cannot be disambiguated and practically used during inference.

**Questions:**

Questions are covered above in the weaknesses section.

---

### Official Review · Reviewer_Dpkt · 2024-11-04

**Soundness:** 3
**Presentation:** 1
**Contribution:** 2
**Rating:** 3
**Confidence:** 3

**Summary:**

The authors present a method for estimating uncertainty in the predictions of deep learning models with high dimensional outputs, with a focus on image data. They suggest that one should estimate the uncertainty in the output space decomposed into two components, corresponding to aleatoric and epistemic uncertainty. They further suggest that these components can be approximated by combining probabilistic predictions across different individual models sampled from some distribution. They propose a method for efficiently estimating output space covariance matrices decomponsed into aleatoric and epistemic components, based on a low rank + diagonal approximation and singular value decomposition. They demonstrate that this method learns data distributions which assign higher likelihood to out of distribution test samples than previously discussed methods, and performs better on an out of distribution detection task. They show how their model performance scales with the number of model samples and SVD components retained.

**Strengths:**

This work significant in that authors suggest a method for estimating uncertainty over high-dimensional outputs of deep neural networks that appears to generate better generative models (as measured by negative log likelihood or pixelwise error), and improve OOD detection relative to existing models benchmarked on the same literature. The particular formulation of this method appears to be novel, in that they estimate aleatoric uncertainty by averaging the covariances derived from individual models, and reduce the rank of the estimated covariance matrix using SVD. Both of these methodological improvements are shown to improve the quality of covariance matrix estimation as measured on a variety of image based downstream tasks, and suggest a fairly general framework for covariance matrix estimation that could be applied to ensemble/Bayesian methods for deep neural networks.

**Weaknesses:**

I have grouped my weakness comments into 2 main groups. Overall, I am concerned that the actual contributions of this paper are not well reflected in its framing, and that more care needs to be taken to align this work with existing efforts to quantify epistemic and aleatoric uncertainty. I am willing to reconsider my score if my main concerns within these groups are addressed.

1. From reading this paper, it is difficult to appreciate what exactly are the novel contributions of this work, and what has been done previously.
    - In particular, from looking at the abstract, introduction, Figure 1, and discussion, my impression would be that the main novel contribution of this work has been to apply the low rank plus diagonal covariance decomposition for the joint estimation of epistemic and aleatoric uncertainty. These sections should be revised to emphasize what is discussed in the related work: namely that 1) the low rank plus diagonal has been fruitfully used for uncertainty estimation in the past (albeit for aleatoric uncertainty mostly), and that there have been other efforts to use this decomposition on both epistemic and aleatoric uncertainty jointly, namely in Zepf et al. 2023 as discussed by the authors. The specific methodological changes that led to improved performance in the results should be emphasized as the novel contribution of this work.

2. Beyond motivation, it is not clear how the epistemic vs. aleatoric uncertainty distinction figures into the results provided.
    - In particular, I think that more care needs to be taken to study how the decomposition of uncertainty, as suggested by equation 1, maps onto the practical properties of the resulting uncertainty components (e.g. how epistemic/aleatoric uncertainty scale with amount of data available). I have made recommendations to address this point in the questions below.
    - More generally, this concern is related to an ongoing discussion in the community regarding the common failure cases of attempts to disentangle epistemic and aleatoric uncertainty (e.g. Wimmer et al. 2023, Mucsanyi et al. 2024). It would be important to discuss how the results of this paper relate to the general challenge of disentangling epistemic and aleatoric uncertainty.

Additionally, I found some details of the presentation to be confusing. I have included clarifying questions in the section below.

References:
- Wimmer, Lisa, et al. "Quantifying aleatoric and epistemic uncertainty in machine learning: Are conditional entropy and mutual information appropriate measures?." Uncertainty in Artificial Intelligence. PMLR, 2023.
- Mucsányi, Bálint, Michael Kirchhof, and Seong Joon Oh. "Benchmarking uncertainty disentanglement: Specialized uncertainties for specialized tasks." arXiv preprint arXiv:2402.19460 (2024).

**Questions:**

- [Related to Weakness 1] Is equation 1 a novel decomposition of epistemic and aleatoric uncertainty? If so, please state so explicitly and discuss the relationship to existing decompositions (e.g. Pfau 2013, Depeweg et al. 2018, Abe et al. 2022, Gupta et al. 2022).
- [Related to Weakness 2] For downstream applications, we want to distinguish between the individual contributions of epistemic and aleatoric uncertainty. To this end, it would be important to consider how the uncertainty components described as epistemic vs. aleatoric here behave as we alter training conditions, in the spirit of the experiments considered by Wimmer et al. 2023. For example:
    - What is the contribution of these individual uncertainties to OOD detection? How do these contributions change scale as we e.g. change the size of the training set? We would hope that the gap between InD and OOD uncertainty is driven largely by epistemic uncertainty, and that this gap grows/shrinks as we consider more/fewer training samples.
    - What if we add noise to the training samples as data augmentation? How are estimates of epistemic vs. aleatoric uncertainty influenced? We would hope that additional noise in the training samples would be reflected by increased aleatoric uncertainty, but not epistemic uncertainty estimates.

- Various points of clarification.
    - Figure 2 is the only place where we actually see definitions for $\mu_{w},P_w$ before they are referenced in main text equations. Please include an explicit discussion of these terms in the text.
    - 304: Why is this section labeled as "Hyperparameter?" "Evaluation" or "Model Specifics" would be more appropriate.
    - The training loss is never defined in the main text. While it is included in section D.3, this appendix section is never referenced in the main text.
    - 378: For the Qualitative results, it would be useful to have samples from competing methods to consider as well, if only in the supplementary methods.
    - 3.5 Ablation Studies: The main result presented here is not an ablation, but rather a hyperparameter sweep. "Effect of hyperparameters" or "Impact of model specifics" would be a more appropriate section titles.

References:
- Pfau, David. "A generalized bias-variance decomposition for bregman divergences." Unpublished manuscript (2013).
- Depeweg, Stefan, et al. "Decomposition of uncertainty in Bayesian deep learning for efficient and risk-sensitive learning." International conference on machine learning. PMLR, 2018.
- Abe, Taiga, et al. "Deep ensembles work, but are they necessary?." Advances in Neural Information Processing Systems 35 (2022): 33646-33660.
- Gupta, Neha, et al. "Ensembles of classifiers: a bias-variance perspective." Transactions on Machine Learning Research (2022).

---

### Note · Authors · 2024-11-28

**Comment:**

We would like to express our gratitude to the reviewers for their thoughtful and constructive feedback.

While we believe that the correct formulation of the joint representation is a critical and nontrivial step towards Bayesian uncertainty estimation with represented correlations in high-dimensional output settings, we recognize the importance of explicitly addressing the separation of epistemic and aleatoric uncertainties. Incorporating this distinction would indeed add significant value to the method.

Unfortunately, evaluating the specific contributions of these uncertainty components proved to be nontrivial, and we were unable to resolve these challenges during the rebuttal period. Therefore, after careful consideration, we have decided to withdraw our current submission.
We plan to further investigate these aspects, including additional analysis on stability, condition numbers, and clarifying ambiguities and imprecise formulations. Once we address these points, we aim to resubmit our improved work to another venue.

Thank you for your valuable insights and support in improving our work.

**Withdrawal Confirmation:**

I have read and agree with the venue's withdrawal policy on behalf of myself and my co-authors.